# Cortical depth profiles in primary visual cortex for illusory and imaginary experiences

Johanna Bergmann ®[1,2,3] ✉, Lucy S. Petro[1,2], Clement Abbatecola ®[1,2], Min S. Li[2,4], A. Tyler Morgan[1,2,5] & Lars Muckli ®[1,2] ✉

Visual illusions and mental imagery are non-physical sensory experiences that involve cortical feedback processing in the primary visual cortex. Using laminar functional magnetic resonance imaging (fMRI) in two studies, we investigate if information about these internal experiences is visible in the activation patterns of different layers of primary visual cortex (V1). We find that imagery content is decodable mainly from deep layers of V1, whereas seemingly 'real' illusory content is decodable mainly from superficial layers. Furthermore, illusory content shares information with perceptual content, whilst imagery content does not generalise to illusory or perceptual information. Together, our results suggest that illusions and imagery, which differ immensely in their subjective experiences, also involve partially distinct early visual microcircuits. However, overlapping microcircuit recruitment might emerge based on the nuanced nature of subjective conscious experience.

During visual illusions, we might: (1) see an object that is not physically present[1]; (2) not see an object that is physically present[2]; or (3) perceive an object's physical properties to be different to how they actually are[3]. Such illusions are attributed to low-level processes, such as cortical feedback from other visual areas[4] or lateral interactions within primary visual cortex[3,5]. For example, during the Kanizsa or the neon colour-spreading illusions, we see illusory contours of a (coloured) shape that is physically not present. This experience might arise because higher-order visual areas with larger receptive fields integrate input from larger portions of the visual field and assume the presence of a shape based on the alignment of other shapes in the visual field[6,7]. Such hypotheses about the global characteristics of individual objects are fed back to lower-order areas, speeding up and facilitating perception[8], but sometimes leading to striking misperceptions.

Illusions, like hallucinations, seem 'real', and our volitional control of their experience is limited. They appear embedded within the external environment, indistinguishable from physical reality. By contrast, everyday mental imagery feels segregated from reality and can influence perception and vice versa[9,10]. In addition, we frequently engage in imagery and perception simultaneously, freely forming visual thoughts that guide our actions and decisions, while simultaneously processing perceptual input from our senses. Evidence suggests a widely distributed network of brain areas is involved in imagery, including recruitment of early visual cortex by high-level frontoparietal regions for the sensory representation of a mental stimulus[11]. We can therefore make predictions based on cortical anatomy about the local connectivity of such top-down input in sensory areas. Most feedforward sensory input from the eyes arrives in the middle layers of the primary visual cortex, whereas most feedback input from other brain areas arrives in the deep and superficial layers[12,13]. On this basis, we hypothesised that information about non-physical visual experiences, like imagery and illusions, should be predominant in the layers that receive feedback from other brain areas. Specifically, cortical models in rodents and monkeys show that

[1]Imaging Centre of Excellence (ICE), Queen Elizabeth University Hospital, University of Glasgow, Glasgow, UK. [2]Centre for Cognitive Neuroimaging (CCNi), School of Psychology and Neuroscience, College of Medical, Veterinary and Life Sciences, University of Glasgow, Glasgow, UK. [3]Department of Psychology, Max Planck Institute for Human Cognitive and Brain Sciences, Leipzig, Germany. [4]Centre for Computational Neuroscience and Cognitive Robotics, School of Psychology, University of Birmingham, Birmingham, UK. [5]Functional MRI Core Facility, National Institute of Mental Health, NIH, Bethesda, MD 20817, USA. ✉e-mail: Bergmann_johanna@yahoo.de; Lars.Muckli@glasgow.ac.uk

long-distance feedback projections from distant brain regions travel through deep layers, with some bifurcations from far-away areas also targeting superficial layers. In contrast, short-range feedback axons that connect to nearby regions are located in and target mostly superficial layers, with some bifurcations also arriving in deep layers[12,14–17]. In light of this, the Dual Counterstream Architecture hypothesis has posited that feedback signals in superficial and deep layers may serve distinct roles in information processing[17]. Therefore, we investigated superficial and deep layers separately to explore how the content of low-level visual illusions, which arise from feedback processing within the visual cortex, and mental imagery, which involves feedback processing from a widely distributed network, is present in these layers.

## Results and discussion
### Experiment 1
We used high-resolution laminar fMRI (0.8 mm³) at 7 Tesla in human participants ($N = 16$) to explore how the content of illusory perception and visual imagery is decodable from different depths of V1. It has been suggested that the source of the fMRI signal originates to a considerable extent from the energy consumption during pre- and post-synaptic dendritic activities, and to a lesser amount from the spiking output at the soma[18]. This means that even when the soma of a pyramidal neuron is located in layer 5, energy consumption may occur elsewhere, for example, in the apical dendrites, which are located in the superficial layers. Using high-resolution laminar fMRI to examine signals at different cortical depths hence holds the potential to provide insights into processes that occur at those specific depths. During the fMRI measurement, participants fixated while we presented five conditions (Fig. 1A, B and 'Methods'): three conditions measured activity patterns during mental imagery, perception, and illusory perception; two conditions acted as controls for the illusory perception condition. During the mental imagery task, we instructed participants to imagine a central red or green disc. In the perceptual condition, participants viewed a central red- or green-coloured disc. During the illusory perception condition, participants viewed the neon colour-spreading illusion, in which four pacman-like ring stimuli induced the illusion that a red or green square shape sits between them (Fig. 1A). In the first control of the illusory perception condition, we presented participants with an 'amodal' version of the stimulus, in which we placed a white contour in the area between the pacman-like rings. This contour breaks or attenuates the illusory experience or can be perceived as overlaying the (illusory) shape behind it. The second control consisted of a mock version of the illusion, in which the coloured quarter of the rings were rotated outwards, such that no illusory shape arises.

Individuals vary in their ability to form mental images. To increase the probability of finding imagery-related signals in V1, we focused on individuals with good imagery abilities. We pre-screened participants with a behavioural task that determines individual imagery strength by measuring its impact on subsequent perception[10] ('Methods' and Supplementary Fig. 1). Only individuals with imagery scores at or above an a priori-defined threshold were invited to participate in the fMRI session.

In experiment 1, we were interested in cortical processes corresponding to two areas of the visual field: (1) the central area around the fixation cross, i.e., the central portion of the area where the participants imagined a coloured disc in the mental imagery task; and (2) four portions in the visual periphery, where the illusory contours of the illusion are located (dashed white lines in Fig. 1C). To identify the V1 portions that process input from these visual field areas, we estimated each voxel's population-receptive field from visual field mapping stimulation[19], and used this approach to define regions of interest (ROI) for further analysis (Supplementary Figs. 2 and 3; 'Methods'). We then segmented the identified V1 regions into

six cortical depth layers (Fig. 1D). For each cortical depth, we used multivariate pattern analysis to determine whether we can detect information about illusory or imagined content. We trained a support vector machine (SVM) classifier to distinguish the red versus green colour of the (illusory/imagined/physical) stimuli in each of the five conditions.

As mental imagery is a cognitive function that involves a distributed cortical network and high-level feedback processing, but no corresponding feedforward input from the eye, mental imagery-related information should be predominant at cortical depths that contain such feedback processing. As participants were instructed to imagine a stimulus centrally, we expected decoding to be above-chance level in foveal V1. This was indeed the case: our SVM classifier was able to decode the imagined colour only in the central ROI; this information was available at the deepest depth ($\hat{\mu} = 0.60$, $P_{adj} = 0.02$, 90% CI [0.54, 0.67], bootstrapped and FDR-corrected; Fig. 2A) but not at any other depth (all $P_{adj} > 0.05$). There was no significant decoding in any of the depth layers in the peripheral V1 region (all $P_{adj} > 0.05$; Fig. 2B), which was at the fringes of the location of imagery.

Perceptual illusions like the neon colour-spreading illusion are thought to arise because higher-order visual areas with larger receptive fields assume the presence of a shape. Feedback signals from nearby visual areas arrive in superficial layers[12]. Previous fMRI research has indeed shown that information about the contextual surround can only be decoded in the superficial layers[20]. However, another study found an increase in fMRI activity levels during illusory perception that was limited to the deeper layers[7]. We only found significant decoding at the second most superficial depth ($\hat{\mu} = 0.59$, $P_{adj} = 0.007$, 90% CI [0.54, 0.64], Fig. 2B); this relationship was confined to the V1 regions representing the areas along the illusory contours, where the illusion is most vivid.

Perception is mediated by both feedforward and feedback processes, and should be decodable across all layers[20]. This was indeed what we found: the physical colour was decodable across all layers in the V1 region that represented the central visual field (all $P_{adj} < 0.02$, Fig. 2A). The same was found for the V1 region that represented the peripheral areas at the fringes of the coloured disc (all $P_{adj} < 0.01$, Fig. 2B).

In the 'amodal' version of the neon colour-spreading illusion, where the illusion is attenuated by a superimposed white contour, we did not find any significant above-chance decoding in either of the ROIs (all $P_{adj} > 0.23$), nor for the mock version of this illusion, where no illusion should arise (all $P_{adj} > 0.3$, Fig. 2A, B, right).

The pattern of significant decoding for experiment 1 seems to reflect the conscious visual experience in space: we could decode imagery only in deep layers in the centre, where our participants imagined the colour. In contrast, illusory colour was not decodable here. However, it was decodable in the superficial layers in the periphery, where the illusory boundaries were located.

### Experiment 2
Our results suggest that cortical feedback related to both low-level illusory perception and high-level mental imagery can be decoded in the layers of V1: mental imagery was decodable in the deep layers of the foveal region, where subjects imagined the colour, and illusory colour was decodable in the superficial layers of the peripheral region, where the illusory boundaries were located. However, a more stringent test would be to examine the decoding of illusory and imaginary feedback signals in the same region of V1. To test this idea and further probe the robustness of our results, we conducted a second fMRI study ($N = 10$). The design was identical except for one important difference: we shifted the Pacman-like ring stimuli of the illusory perception condition and its 'amodal' and mock version sideways, such that one of the illusory boundaries would be located centrally, just like the perceptual and mental imagery stimulus

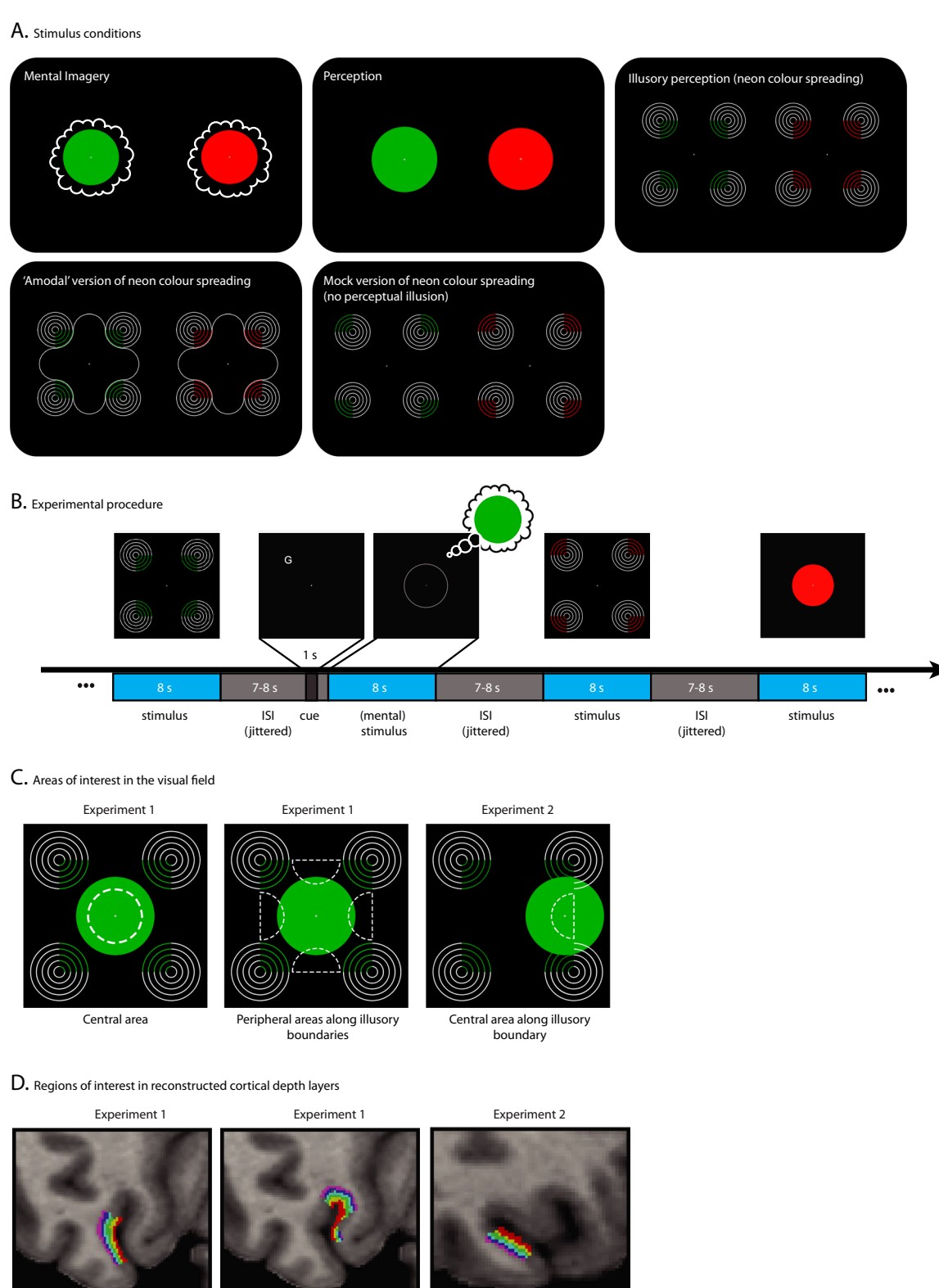

A. Stimulus conditions

Mental Imagery

Perception

Illusory perception (neon colour spreading)

'Amodal' version of neon colour spreading

Mock version of neon colour spreading
(no perceptual illusion)

B. Experimental procedure

G

1 s

| 8 s | 7-8 s | cue | 8 s | 7-8 s | 8 s | 7-8 s | 8 s |

stimulus | ISI (jittered) | | (mental) stimulus | ISI (jittered) | stimulus | ISI (jittered) | stimulus

C. Areas of interest in the visual field

Experiment 1 — Central area

Experiment 1 — Peripheral areas along illusory boundaries

Experiment 2 — Central area along illusory boundary

D. Regions of interest in reconstructed cortical depth layers

Experiment 1 — Central

Experiment 1 — Peripheral

Experiment 2 — Central

(Fig. 1C). With this modification, the colours of the illusory contour and the imagined colours of the mental imagery condition should now both be decodable from the same region-of-interest. This was indeed the case: Mental imagery was again decodable from deep cortical depths ($\hat{\mu} = 0.6$, $P_{adj} < 0.001$, 90% CI [0.55, 0.65] at the deepest depth and $\hat{\mu} = 0.6$, $P_{adj} = 0.03$, 90% CI [0.53, 0.67] at the second deepest depth; Fig. 2C). In contrast, illusory perception only showed significant above-chance decoding in the second most superficial depth layer ($\hat{\mu} = 0.59$, $P_{adj} = 0.03$, 90% CI [0.53, 0.66], Fig. 2C). At the more foveal ROI, the 'amodal' condition now showed above chance decoding in the most superficial layer ($\hat{\mu} = 0.56$, $P_{adj} = 0.03$, 90% CI [0.52, 0.60]). In separate experiments, we have observed that occluded natural scene information is found in superficial layers[20].

**Fig. 1 | Experimental stimulation and procedure, visual areas of interest and cortical depth-layer segmentation. A** Five experimental conditions were used. The stimuli were in red or green to enable colour to be decoded within each condition. **B** Stimuli were presented in a randomised order. During mental imagery, a faint grey circle indicated the location of where to imagine the colour. A letter cue preceding the imagery phase indicated which colour to imagine. It was shown at random locations outside of the field of imagery, with the distance to the fixation cross held constant. **C** V1 regions of interest (ROI) were based on the visual field portions they represented (dashed white lines). Note that the pacman-like rings of the illusory perception condition and the central colour disc were never presented at the same time and are only displayed here together for illustrative purposes. In the second experiment, the stimuli of the illusory perception condition and its 'amodal' and mock version were shifted sideways. This way, one illusory boundary was located centrally, thereby overlapping with the (internal) stimuli in the perception and mental imagery condition. **D** In each V1 ROI, the grey matter was segmented into 6 cortical depth layers, which are partially overlapping and equally spaced, and therefore do not represent anatomical layers that are unequally distributed; neighbouring layers may hence partly share the same voxels. Shown here are the right hemisphere portions of the ROIs.

## Second-level analysis across both experiments

In a second-level statistical analysis across the two experiments, we defined a linear mixed effects model parameterised across cortical depth to more directly compare mental imagery and illusory perception decoding at the different cortical depths, and to examine any statistical differences between the two experiments. To do so, we pooled the mental imagery decoding data from the central ROI and the perceptual illusory decoding data from the peripheral ROI from the first experiment with the mental imagery and illusory perception decoding data from the central ROI of the second experiment (Fig. 3). We found a significant interaction between condition and depth ($t(283) = 3.70$, $P < 0.001$, beta = 0.34, 95% CI [0.16, 0.52]). This was true while controlling for the effect of experiment 1 vs. 2, as well as random effects over participants. There was a significant main effect of depth ($t(283) = -2.13$, $P = 0.034$, beta = −0.16, 95% CI [−0.32, −0.01]) and of condition ($t(283) = -5.10$, $P < 0.001$, beta = −0.19, 95% CI [−0.26, −0.12]), as well as a significant interaction effect between experiment and condition (more accuracy for mental imagery in the second experiment, $t(283) = 3.22$, $P = 0.001$, beta = 0.19, 95% CI [0.07, 0.31]). There was no significant main effect of experiment ($t(283) = -1.35$, $P = 0.178$, beta = -0.07, 95% CI [−0.17, 0.03]) and no significant interactions between experiment and depth ($t(283) = 1.47$, $P = 0.143$, beta = 0.14, 95% CI [−0.05, 0.33]), or between experiment, condition and depth ($t(283) = -1.43$, $P = 0.155$, beta = −0.17, 95% CI [−0.40, 0.06]), together indicating that the experiment did not significantly affect the slope. In conclusion, the interaction between condition and cortical depth indicates that decoding significantly differed between the critical experimental conditions (mental imagery and illusory perception) at the different cortical depths. This analysis complements the first-level bootstrapping analyses showing that illusory content is significantly decoded only in the superficial cortical depths, while imagery content is significantly decoded only at deeper cortical depths. However, the absence of significant decoding of imagery in the superficial layers is not evidence for no decoding in superficial layers. Note that we did see some individual participants with decoding of imagery content in the superficial layers (especially in experiment 2, but also in some participants in experiment 1, Supplementary Figs. 10 and 12). Further investigations might be able to link decoding fluctuations to the subjective strength of imagery experienced (see discussion below).

## Additional decoding and cross-classification analyses

Although this superficial layer multivariate decoding effect for illusory perception contradicts earlier laminar fMRI findings using univariate approaches[7], it aligns well with results on edge perception between figure and ground from electrophysiology research in mice[21] and primates[22], and the finding that superficial V1 layers in mice respond to illusory contours in the Kanizsa illusion[23]. It should be noted that we use colour stimuli, whereas the earlier laminar fMRI findings used orientation stimuli[7]. Whether and how this may have influenced the results remains an open question. Neurons responding to orientation may be distributed throughout the column. However, colour processing, too, appears to involve many layers. Colour blobs are present in upper layers, while feedforward and feedback signals that carry red-green colour information are also processed in mid and deep layers[24].

Additional analyses provide further support to our finding that superficial layers carry illusory stimulus information: when decoding the illusory stimulus against its mock version, consistent significant above-chance level decoding was only present at superficial depths (Supplementary Fig. 4 and 'Methods'). The modified design of the second experiment also allowed us to conduct another analysis: as the illusory boundaries of the illusory perception condition were now located in the same portion of the visual field as the imagined and the perceptual stimuli, we could conduct a cross-classification analysis: we trained the SVM classifier on the two colours of one condition and tested it on another. Interestingly, when training the classifier on illusory colour and testing it on perceptual colour, we found that significant above-chance decoding was again only present in superficial layers, suggesting that information between illusory and actual perception is shared at these depths (Supplementary Fig. 5). In contrast, we could neither cross-classify between mental imagery and illusory perception, nor between mental imagery and perception, irrespective of whether we trained on mental imagery and tested on (illusory) perception or vice versa. This suggests that V1 feedback information of mental imagery may be more distinct. However, it has been shown previously that a classifier trained on response patterns when participants saw oriented gratings could be successfully tested on response patterns when participants were mentally rotating the gratings[25]. This discrepancy requires follow-up investigations but could plausibly relate to the fact that the processing of static imagery, as in our experiment, might be different to the mental rotation task used previously. Evidence in support of this hypothesis comes from congenital aphantasia where participants who are unable to form visual mental images perform with similar accuracy as controls in mental rotation tasks[26].

## Laminar decoding accuracy and BOLD amplitude appear uncorrelated

Due to larger blood vessels at the pial surface of the cortex, the signal of gradient-echo fMRI imaging is stronger in superficial depths[27]. This effect was also present in both of our data sets (Supplementary Fig. 6–9). One could argue that SVM classification performance could be influenced by this, in the way that decoding accuracy could be higher when the overall signal—and hence the signal-to-noise ratio—is enhanced. Alternatively, the fMRI signal might also be more spatially specific when the point spread of the blood-oxygen-level-dependent (BOLD) signal is smaller due to smaller vessels at deeper depths[28]. This, in turn, could lead to better decoding at deeper depths. The pattern of our results, however, does not suggest that SVM classification is dictated by overall BOLD activity levels at different cortical depths: we found that imagery was only decodable in the deep layers, where the signal strength was the lowest; conversely, illusory perception was only decodable in the superficial layers, where the signal was highest; and the perception of physical stimuli was decodable at all depths, regardless of different signal strengths. Decoding accuracy and amplitude hence appear uncorrelated from one another, which is consistent with previous findings[20].

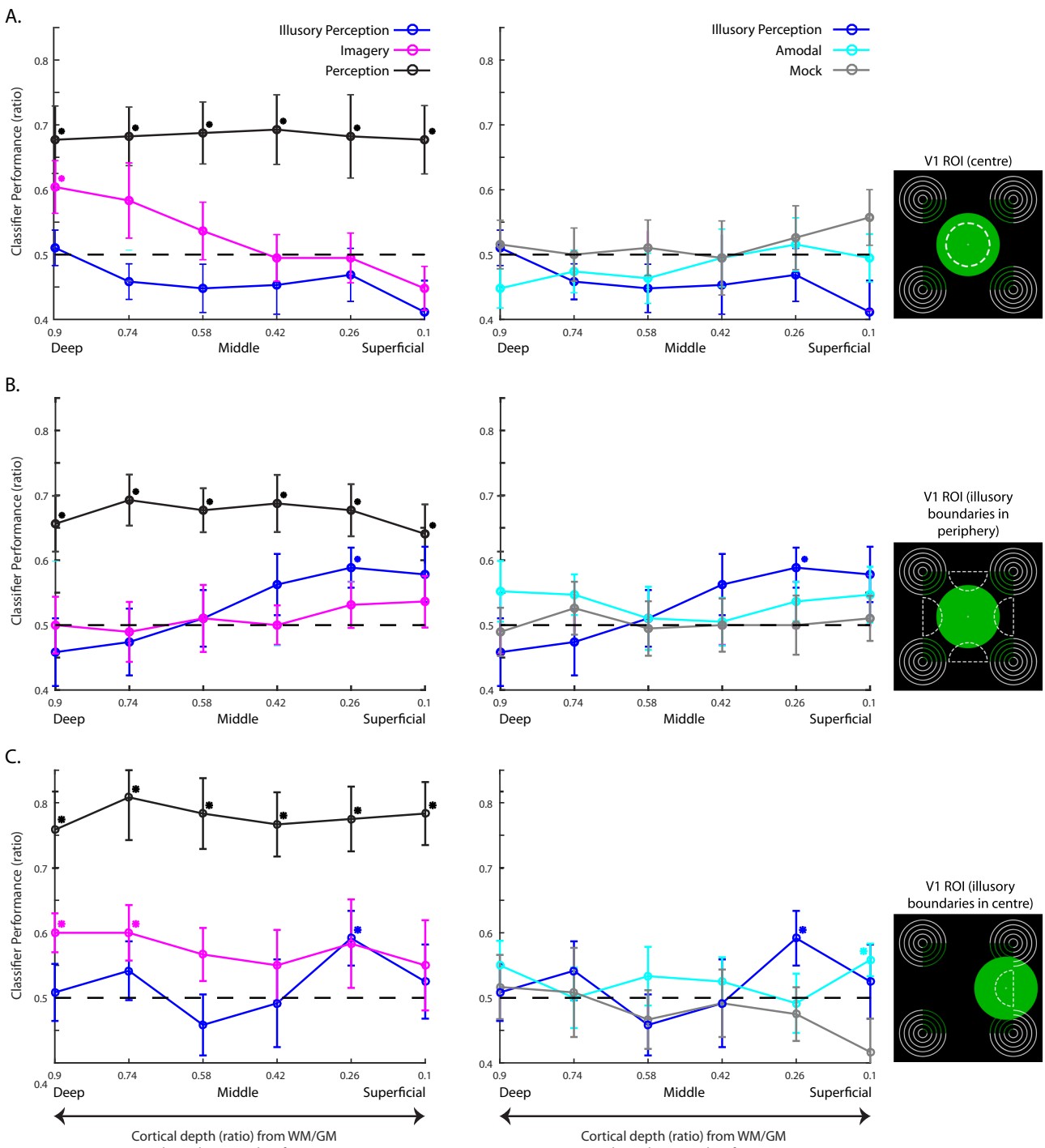

**Fig. 2 | SVM decoding results.** Results for experiment 1 ($n = 16$) are shown in (**A**, **B**); experiment 2 ($n = 10$) results are shown in (**C**). The white dashed lines in the images on the right depict the visual field areas that were used to define the ROIs in V1 from which we decoded. The black dashed lines in the plots represent chance level; asterisks denote significant above-chance decoding ($P_{adj} < 0.05$, one-sided bootstrapping of the mean, multiple-comparison corrected); error bars represent ± SEM. The left column denotes our main conditions of interest: perception (black), illusory perception (blue) and imagery (pink). The right column denotes our illusory control conditions: amodal (cyan) and mock (grey), benchmarked against illusory perception (blue). See Supplementary Figs. 10–12 for individual subject plots. When participants imagined a central disc, colour was decodable only at deep depths in both experiments. Importantly, these findings were restricted to the ROIs that represented the visual area where participants imagined the stimuli (**A**, **C**, left); no significant decoding was found in the peripheral ROI of experiment 1

(**B**, left), which lay at the fringes of the imagery field. Conversely, illusory colour was only decodable in the second superficial layers of the peripheral ROI of experiment 1 (**B**), which represented areas along the illusory boundaries of the neon colour-spreading stimulus. However, when the illusory contour was shifted to a central location in experiment 2, the illusory colour was also decodable in the second superficial layer of the foveal ROI (**C**). Hence, experiment 2 shows that the deep-layer effect for imagery and the superficial-layer effect for illusory perception were detectable within the same ROI. In the perceptual condition, we could significantly decode colour at all depths and in all ROIs of the two experiments. The two control conditions of the illusory stimulus showed no significant decoding in either of the two ROIs in experiment 1. In experiment 2, the 'amodal', i.e., occluded version of the illusory stimulus showed significant decoding in the most superficial layer ($\hat{\mu} = 0.56$, $P_{adj} = 0.03$, 90% CI [0.52, 0.60]). Source data are provided as a Source Data file.

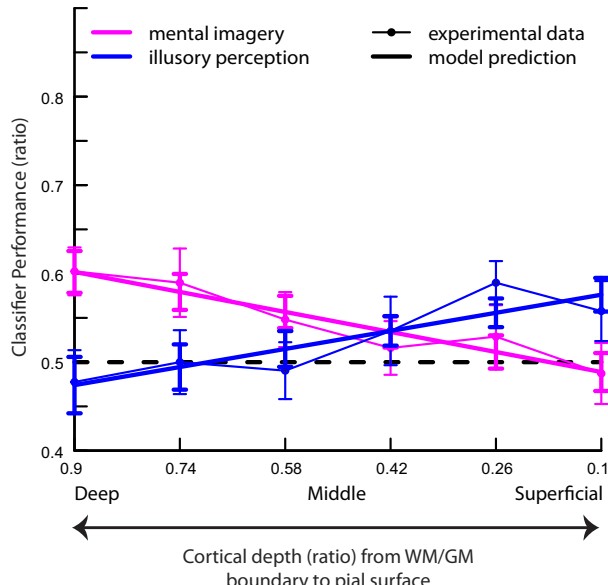

**Fig. 3 | Second-level analysis of the two critical stimulus conditions.** In a second-level analysis across both experiments, we fitted a linear mixed model to predict decoding accuracy with depth, experiment and the two critical stimulus conditions (mental imagery and illusory perception). Model predictions for both conditions (across experiments; thicker lines) are plotted alongside the experimental data (thinner lines). For separate model predictions for the individual experiments, see Supplementary Fig. 13. Source data are provided as a Source Data file.

### The functional roles of superficial and deep cortical layers

Laminar fMRI provides new ways of identifying processing characteristics in overlapping cognitive and perceptual functions at a level of precision that has been unattainable with conventional fMRI. Information in deep and superficial layers may be indicative of different sources of feedback in the brain, with far-distant sources sending the majority of their input to deep layers, and nearby sources sending most of their signals to superficial layers of V1[12,14,17]. In this context, an intriguing series of questions arise: how do different types of laminar feedback shape our conscious experience? Is low-level feedback to superficial layers linked to high-precision perceptual or (almost) perception-like experiences? Does feedback to deep layers serve the formation of a more malleable 'inner' sensory world that we can keep separate from our ongoing perception? Does feedback information in superficial layers reflect a negative prediction error related to the expectation of a contour when in fact no contour is present? Interestingly, although overall decoding of mental imagery remained non-significant in superficial layers, there were some individuals in our experiments who showed above-chance decoding here too, whereas others did not (Supplementary Figs. 10 and 12). It is possible that inter- (and intra-) individual differences in the laminar information profile of mental imagery might account for how strong or 'real' it might be experienced: that is, a mental imagery experience might appear stronger, more precise or more 'real' when a larger portion of imagery-related signals reach not only deep, but also superficial layers via the longer/slower path through the visual cortex hierarchy, or through bifurcations from the deep layers. On the other hand, one study found that visual expectations—possibly arising from feedback from the hippocampus—only led to activity increases in deep layers[29]. As strong visual expectations or predictions are also thought to be the basis of hallucinatory experiences, this would speak against the idea that the 'realness' of experiences is necessarily linked to superficial layer feedback signals. We only used a sample of subjects with high imagery ability, so future research is needed to compare variability in imagery strength and its correlations with brain data.

In this work, we find that imagery content is decodable mainly from deep layers of V1, whereas seemingly 'real' illusory content is decodable mainly from V1 superficial layers. More research is necessary to disentangle whether our ability to exert volitional control over conscious sensory experiences and our perception of 'realness' are linked, or independent of each other. A recent study proposed that internal stimulation can be perceived as real once it passes a reality threshold[30]. Our finding that unlike perception, mental imagery is only significantly decodable in certain V1 layers, and that illusory perception and perception share information in V1, whereas mental imagery does not, will further help us to refine theories that mental imagery is 'perception-like' in nature[11]. This important difference in layer-wise processing had been missed with conventional fMRI technology, which lacks the precision to look at brain signals at different cortical depths.

## Methods

### Participants

fMRI data were collected from 18 healthy participants in Experiment 1 (19–36 years, $M = 23.3$, SD = 4.96; 14 females and 4 males), and 12 healthy participants in Experiment 2 (19–31 years, $M = 23.2$, SD = 4.02; 8 females and 4 males;). All had normal or corrected-to-normal vision. The participants were recruited from a larger sample who underwent a behavioural pre-test designed to quantify individual imagery strength. This pre-test sample consisted of 55 individuals in Experiment 1 (18–36 years; 20 males, 35 females) and 52 individuals in Experiment 2 (18–33 years, $M = 23.1$, SD = 3.76, 18 males, 34 females). Four of the participants of Experiment 2 had already participated in Experiment 1, and therefore did not do the pre-test again. Of the pre-sample of Experiment 1, 21 participants had an imagery strength score that met or surpassed an a priori-defined threshold and were invited to take part in the main fMRI experiment. In total, 18 of these completed the fMRI experiment. For Experiment 2, 11 participants surpassed the a priori-defined threshold (plus the 4 participants who had already taken part in Experiment 1, i.e., 15 in total). In all, 12 of them completed the fMRI experiment. Due to bad inter-run alignment in two data sets in Experiment 1 and two data sets in Experiment 2 (see functional GE-EPI data pre-processing), analyses were conducted with the data sets of 16 participants (Experiment 1), and 10 participants (Experiment 2). Participants were recruited from the participant pool of the Institute of Neuroscience and Psychology, University of Glasgow. Preceding the study, fMRI piloting sessions were conducted with 3 participants at the Maastricht Brain Imaging Centre, Netherlands. All subjects gave informed consent and were paid for their participation. The study was approved by the ethics committee of the College of Science and Engineering and the College of Medical, Veterinary and Life Sciences of the University of Glasgow.

### Behavioural pre-test to quantify individual imagery strength

The ability to imagine visual content varies greatly between individuals[31–34]. Furthermore, an individual's ability to form a mental image is associated with enhanced decoding accuracy of fMRI activity patterns in early visual cortex[35]. This suggests a link between mental imagery ability and the precision of imagery-related visual cortex signals measured by fMRI. To increase the probability of finding meaningful imagery-related V1 activity patterns and thereby increase the statistical power of the study[36], we ran behavioural pre-screenings to identify individuals with good visual imagery abilities. We used a behavioural paradigm that quantifies individual imagery strength by measuring its impact on subsequent conscious perception of a binocular rivalry display[10,32–34]. Binocular rivalry occurs when the two eyes see two different images, one to each eye. This setup results in the phenomenon that perception alternates between the two images, with one image reaching conscious awareness while the other one is suppressed[37,38]. Importantly, previous work has shown that preceding

imagery can bias brief subsequent binocular rivalry perception, with the previously imagined stimulus having a higher chance to gain dominance. This bias is further increased with longer periods of imagery[10], and when the stimulus is imagined more vividly in a given trial[34,39]. In addition, individual imagery ability quantified by this paradigm is linked to individual visual working memory storage[32,33], and is highly stable over time[34]. Taken together, evidence suggests that determining individual imagery ability by quantifying its impact on subsequent binocular rivalry perception might be a valid and reliable method with which to identify individuals with good imagery ability.

In the behavioural pre-test session, participants sat in a darkened room at a distance of 60 cm from a computer screen (ASUS PG278Q, 27"), their heads stabilised using a chin rest. They wore crystallised shutter glasses (NVIDIA 3D VISION 2), which allows different images to be projected separately to each eye, thereby making it possible to induce binocular rivalry. Experimental stimulation was controlled using MATLAB R2016a, in combination with the Psychtoolbox v3.0.13 extension[40], running on a DELL Precision T3500 computer with an Intel® Xeon® CPU processor and an NVIDIA GeForce GTX 970 graphics card. Participants were instructed to maintain fixation on the central fixation cross throughout the experiment. Apart from the white fixation cross, the screen remained black. Participants completed two tasks. Prior to the imagery strength task, participants did an eye dominance task designed to adjust the luminance of the two stimuli to each individual's eye dominance. Individual differences in eye dominance would otherwise affect binocular rivalry perception, resulting in a bias for the image that is viewed with the more dominant eye. The procedure has been described previously[10,34]; in short, it applies an adaptive testing procedure designed to individually adjust the luminance of the two colours such that it is equally likely for each to gain perceptual dominance. Following the eye dominance task, participants completed two training trials of the imagery test to familiarise themselves with the task. If necessary, further adjustments of the luminance values were done before the main imagery task.

At the beginning of each trial of the imagery strength task (Supplementary Fig. 1A), a grey letter 'R' or 'G' (1.3 cm in size, i.e., 1.2° visual angle) appeared in the central lower part of the screen, at a distance of 4.9 cm (4.7°) from the fixation cross. The letter cued the participants as to which of the two colours they should imagine in the given trial—'R' for red, 'G' for green. The cue was shown for 1 s, and was followed by a 1 s break, during which only the fixation cross was visible. Following this phase, a faint grey circle (radius of 3.65 cm; 3.5°) was presented for 7 s, centred on the fixation cross. During these 7 s, participants were asked to imagine the colour as vividly as possible and within the frames of the faint grey circle on the screen. Following this 7-s imagery phase, the circle disappeared, and the word 'vividness?' appeared in the central lower part of the screen (at a distance of 4.9 cm/4.7° from the fixation cross), instructing participants to rate how vivid their mental image of the colour had been (scale ranging from 1 to 4 where 1 represented the lowest and 4 the highest level of vividness). After responding with a key press, the binocular rivalry display appeared for 0.75 s, showing circular Gaussian-windowed green and red colour stimuli, one shown to each eye. When the binocular rivalry display disappeared, participants were instructed to indicate via a key press which of the two colours had been dominant ('1' for red, '2' green, '3' for mixed). 10% of the trials were catch trials, where a mixture of both colours was presented to each eye; in these trials, participants should give a 'mixed' response; the failure to do so is an indicator of decision bias. Each participant completed one run with $n = 100$ trials, with one break after completing half of the trials. Individual imagery strength was defined as the individual bias with which the imagined stimulus gained perceptual dominance during subsequent binocular rivalry: imagery strength (%primed) = $n_{primed}/(n-n_{mock}-n_{mixed})$, with $n_{primed}$ being the number of (non-catch) trials in which the imagined colour matched the colour that was subsequently dominant during binocular

rivalry, n the total number of trials, $n_{mock}$ the number of catch trials, and $n_{mixed}$ the number of (non-catch) trials in which participants reported a mixed percept. An arbitrary, a priori-defined threshold of 60% priming was set as a threshold to identify individuals with good enough imagery ability (Supplementary Fig. 1B). Individuals that scored priming values at or above the threshold level were invited to participate in the fMRI study. The subjective vividness ratings that participants gave were not used as a criterion for participation in the fMRI sessions but were used in further analyses to check the validity of the pre-test (Supplementary Fig. 1C, D).

## fMRI experimental procedure and design
Unless otherwise stated, fMRI experimental procedure, design, data collection, pre-processing and analysis was done analogously in Experiment 1 and 2. In a 7 T MRI scanner, participants completed 6 runs of task-related experimental stimulation, and 2 retinotopic mapping runs. 2 participants of Experiment 2 completed the retinotopic maps on another day. Throughout the experiment, participants were asked to maintain fixation on the white central fixation cross on an otherwise black screen. Fixation was monitored using an eye tracker (EyeLink 1000 Plus, SR Research). However, the eye tracker was often not able to track the eyes accurately throughout the runs because the head coil limited the camera's field of view. This issue rendered the data unusable for most participants; the data of those participants in which eye tracking was possible throughout all runs suggest that they were able to maintain strong fixation and that eye gaze position correlates very highly across conditions and colours (Supplementary Fig. 14). The screen was attached to the top end of the scanning table and viewed through a mirror attached to the head coil, at a viewing distance of ~96 cm. The task consisted of five conditions, in which participants were asked to view or imagine coloured shapes that were either red or green (Fig. 1A). Each condition was shown in blocks of 8 s, interleaved by inter-stimulus intervals (ISIs) that were jittered between 7 and 8 s. In each run, each condition was shown 6 times, i.e., three times with red stimuli, and three times with green stimuli. The luminance of the stimuli was kept the same for all participants. In the perception condition, participants viewed a uniformly red or green disk presented at the centre (radius of 3.1°; in 4 participants, the radius was 3.7°), flashing at a frequency of 4 Hz. In the imagery condition, participants were asked to imagine that same red or green disc, with the area of imagery indicated by a faint grey circle around the fixation cross. The letter 'R' or 'G' cued participants as to which colour to imagine (red or green). It was presented for 1 s, 2 s prior to the presentation of the faint grey circle, which signalled the onset of the imagery task. Importantly, the cue was presented in the periphery outside of the grey circle presented subsequently, and its location on the screen was randomised but kept at a stable distance of 4.3° to the central fixation cross (5.1° in 4 participants). Since some participants had difficulty seeing the lower central part of the screen due to the head coil, this part of the screen was spared (i.e., the cue never appeared here). In the illusory perception condition of Experiment 1, participants viewed a visual illusion, known as neon colour spreading, in which 4 peripheral Pacman-shaped stimuli elicit the illusion of a coloured (red or green) square, with illusory coloured boundaries continuing beyond the edges of the coloured Pacman shapes in the periphery. The four pacman-shaped stimuli were each presented in one of the four quadrants of the screen, respectively, with the centres of the stimuli located at a distance of 6.24° to the central fixation cross (7° in 4 participants). Each consisted of five concentric rings of increasing radii, the distance between radii remaining constant. The outermost (and largest) ring's closest distance to the central fixation cross was 3.94° (4.71° in 4 participants). The inwardly directed quarter of the rings was either red or green, thereby leading to the illusion of a coloured square. The remaining three-quarters of the ring were white. In the third condition, we presented an 'amodal' version of this illusion, in which a white contour was

placed in the area between the Pacman-like rings, thereby attenuating the illusion (the contour's closest distance to the central fixation cross being 3.86°; 4.61° in four participants). In the mock version of the illusory perception condition, the pacmen were rotated outwards, such that no illusion of a coloured shape should arise. In Experiment 2, which presented a conceptual replication of Experiment 1, we used the same five conditions. However, in contrast to Experiment 1, the Pacman-like stimuli of the illusory, amodal and mock condition were shifted sideways, such that the illusory contour of the right (illusory) edge would fall in the central visual field. This way, the illusory contour should also be represented by the foveal region of V1, as in the perception and mental imagery conditions (which remained the same as in Experiment 1). The outermost (and largest) ring's closest distance of the Pacman-like rings in the lower and upper central region to the central fixation cross was 2.2°. Before starting the experiment, we ensured that each subject was able to see the illusion-inducing peripheral stimuli. Our experience with Experiment 1 showed that some participants had difficulty seeing the lower central part of the screen due to the head coil (in Experiment 1, there were no stimuli shown here). Therefore, in Experiment 2, we ensured that participants' heads were positioned in the head coil in such a way that they would see the lower and upper central parts of the screen, where two of the Pacmen of the shifted stimulus conditions were now located. At the end of each run in both experiments, we also presented contrast-inverting (4 Hz) chequerboard patterns. In Experiment 1, one was a disc in the central target region with a radius of 2.1° (2.6° in 4 participants) from the central fixation cross, and one was ring-shaped, surrounding the central target region (spanning an area from a radius of 2.1° to 4.4° from central fixation; 2.6° to 4.9° in 4 participants). In Experiment 2, the chequerboard target stimulus had the shape of a semi-circle, located along the illusory contour that passed through the central region of the visual field. The area of the semi-circle encompassed the central fixation cross; its straight edge had a total length of 3.6° (i.e., 1.8° upwards and downwards from the horizontal meridian), and passed the central fixation cross to the right at a distance of 0.7° at its closest point. The surround was mapped with a circle-shaped chequerboard that spared the portion of the semi-circle target region. The target > surround contrast was used to map the central target region in V1, and to compare this region with the central target region identified by population-receptive field (pRF) mapping to visually cross-check the validity of the pRF mapping (see below). The ROI definition was then made on the basis of pRF mapping. Note that this central target region identified by the target vs. surround contrast and by the pRF mapping was smaller than the area in which the stimulus was viewed or imagined in the perception and imagery condition. We did this for two reasons: first, to ensure that even with small shifts of the centre of gaze due to unavoidable saccades, the coloured disc would still be represented by foveal V1 neurons. Second, it limited the risk that despite small gaze shifts, the peripheral pacman stimuli of the illusory and control conditions would fall within the receptive fields of the foveal V1 region that we decoded from, as the pacmen contained actual (and not just illusory) colour. Following the six task runs, we also acquired Polar Angle and Eccentricity maps. These data were used: (1) to map V1 retinotopically on the cortical surface[41,42]; and (2) to compute pRF maps, to determine which portions of V1 represent which portions of the visual field[19]. In the polar angle mapping run, participants viewed a contrast-inverting (5 Hz) black and white chequerboard wedge rotating anti-clockwise around the central fixation cross. The wedge had an angle of 22.5° and rotated at a constant angular speed around the fixation cross 12 times, each cycle lasting 64 s. In the Eccentricity mapping run, participants viewed a contrast-inverting black and white chequerboard ring, slowly expanding from the centre of gaze to the periphery. An expansion cycle lasted 64 s and was repeated 8 times. The Polar Angle and Eccentricity measurement induces phase-encoded neural activity that allows us to estimate the boundaries between early visual areas (Polar Angle) and to match the different expansion radii of the visual stimulus to the eccentricity radii on the cortical surface (Eccentricity mapping). Both wedge and ring stimuli were presented in front of a grey background with a target spanning the screen, centred on the fixation cross. Before and after the presentation of the wedge or the ring, there was a baseline phase of 12 s, during which only the background and fixation cross were presented. In both polar angle and eccentricity runs, the participants' task was to maintain fixation on the central fixation cross.

## fMRI data acquisition

fMRI data were acquired in a Siemens 7-Tesla Terra Magnetom MRI scanner with a 32-channel head coil located at the Imaging Centre of Excellence of the University of Glasgow. Each of the six task-related runs consisted of 272 T2*-weighted gradient-echo echoplanar (EPI) images using the CMRR MB sequence with an MB factor of 1 (voxel resolution: $0.8 \times 0.8 \times 0.8$ mm³ isotropic resolution, distance factor: 0%, 27 slices, FoV=148 mm, TR = 2000 ms, TE = 26.4 ms, flip angle: 70°, slice timing: interleaved, bandwidth = 1034 Hz/px, phase-encoding direction: head to foot). The EPI slab was positioned along the calcarine sulcus of occipital cortex, where V1 is located. We used the same sequence for the retinotopic mapping runs, acquiring 396 volumes for the Polar Angle and 268 volumes for the Eccentricity mappings. In order to identify susceptibility-induced distortions, which can distort the functional EPI images and make it hard to align them to the anatomical images, we also recorded five volumes of the same EPI sequence with the phase-encoding direction inverted. This way, it is possible to estimate field distortions and correct for them using FSL's topup tool[43,44]. Note, however, that distortion correction can introduce blurring, and thus introduce the risk that the true laminar activity (pattern) profile remains concealed[45]. It should therefore be used with caution and only if necessary. In addition, whole-brain, high-resolution T1-weighted MP2RAGE images were acquired (voxel resolution: $0.63 \times 0.63 \times 0.63$ mm³ isotropic resolution, 256 sagittal slices, TR = 4680 ms, TE = 2.09 ms, $TI_1$ = 840 ms, $TI_2$ = 2370 ms, flip angle₁ = 5°, flip angle₂ = 6°, bandwidth = 250 Hz/px, acceleration factor = 3 in primary phase-encoding direction, FOV = 240 mm). In order to scale the applied voltage to achieve accurate flip angles for the scanning session, we also ran a 3DREAM B1 mapping sequence in Experiment 1 beforehand[46] (voxel resolution: $4.5 \times 4.5 \times 4.5$ mm³ isotropic resolution, 44 slices, FOV = 288 mm, TR = 5000 ms, $TE_1$ = 0.9 ms, $TE_2$ = 1.54 ms, flip angle₁: 60°, flip angle₂: 8°).

## functional GE-EPI data pre-processing

Functional imaging data were analysed using BrainVoyager 20.6[47,48]. Functional image pre-processing involved 3D-motion correction, slice time correction, high-pass filtering, and coregistration to the T1-weighted anatomical images using boundary-based registration. To ensure the functional runs were excellently aligned to each other, which is imperative for meaningful multivariate pattern analyses with high-resolution voxels, we computed inter-run spatial cross-correlations of the signal intensities of the functional volumes. Where necessary, BrainVoyager's VTC-VTC grid search alignment and spatial transformation tools with sinc interpolation were used to improve inter-run alignment until it was at least $r > 0.9$ on average, and showed good alignment visually. Further, functional-anatomical alignments were checked visually to ensure that the functional scans were well aligned to the anatomical image at the location of and around the ROI (Supplementary Fig. 15). Due to the relatively small ROIs, no susceptibility-induced distortion correction using FSL's topup tool was necessary, thereby avoiding the previously mentioned confounds that distortion correction can cause[45]. However, the data of 2 of the 18 participants in Experiment 1 showed very bad inter-run alignment ($r = 0.13$ and $r = 0.38$, respectively), and after several failed attempts to improve the alignment, the two data sets were removed

from further analysis. Similarly, in Experiment 2, 2 of the 12 participants showed below-threshold inter-run alignment, and one of these two also showed no discernible retinotopic maps in either of the two hemispheres. Although the initial alignment was not as low as that of the two participants in Experiment 1 ($r = 0.72$ and $r = 0.85$), multiple attempts to improve alignment enough to pass the a priori-defined threshold of $r > 0.9$ failed, and therefore the two data sets were removed from further analysis.

## T1-weighted anatomical data pre-processing

We first processed the anatomical imaging data with a range of brain imaging software, before converting them to BrainVoyager format to continue processing. We used FSL to upsample the anatomical data to 0.4 mm³ (FMRIB's Software Library[49], www.fmrib.ox.ac.uk/fsl). For a preliminary white matter-grey matter segmentation, we then processed anatomical imaging data using a custom-written pipeline described previously[50,51]. The pipeline uses R[52], AFNI[53], and nighres[54]. Where necessary, we corrected the results of the automatic segmentation manually using ITK-SNAP[55]. Our focus of the manual corrections was on V1, and we took great care to ensure that the grey matter segmentation result did not contain any parts from the sinus or skull. The corrected segmentation was then used to mask the original images using FSL. We then converted the masked anatomical image, as well as the white matter mask, to BrainVoyager format to continue processing in BrainVoyager. After a rigid-body transformation of the anatomical images into ACPC space, the white matter mask was drawn onto the anatomical image in a first step, before using BrainVoyager's advanced segmentation tool for the GM-CSF segmentation. From the segmented white matter-grey matter boundary, we reconstructed the surface of the occipital lobe. The retinotopic mapping-based estimation of V1 boundaries and the definition of the ROIs within V1 was then done on the inflated cortical surface (see below). This process was done individually for every dataset.

## Retinotopic mapping of V1 boundaries

We estimated the boundaries of V1 using the fMRI data recorded during the previously described retinotopic mapping scans. In order to estimate V1 functionally, a Fourier transform was applied to each voxel's fMRI time series of the polar angle and eccentricity mapping run in order to compute amplitude and phase at stimulation frequency[41,42]. The different phase angles were then colour-encoded and mapped onto the inflated cortical surface. Each colour represented an F-ratio of the squared amplitude divided by the average squared amplitudes at all other frequencies. On the basis of the colour encoding, the boundary segregating V1 from V2 was then estimated manually on the cortical surface for each individual subject. Due to the small slab of the functional volume, only V1 could be estimated. In 2 of the 16 participants, the colour encoding did not reveal a retinotopic map in one of the hemispheres (left hemisphere in S12 and right hemisphere in S13), and therefore V1 could only be estimated in the other hemisphere.

## Regions of interest definition in V1 using population-receptive field mapping (pRFs)

Following the retinotopic mapping-based estimation of V1, we identified our regions of interest in V1 using an approach informed by pRF mapping (Supplementary Fig. 2). Using visual field mapping stimulation, population-receptive field mapping allows to estimate which portion of the visual field an fMRI voxel is most responsive to[19]. To compute pRFs, we first created model time courses that predicted how a voxel's time course responsive to a certain portion of the visual field would look during eccentricity and polar angle mapping stimulation. We assumed a standard isotropic Gaussian model; we first created a Gaussian window for every portion of the visual field, defined by its spatial coordinates $x$ and $y$, as well as by its size (i.e., standard deviation

in the Gaussian model). Models of 24 different sizes (i.e., standard deviations) were estimated, with the centre of each Gaussian window being one standard deviation apart from the next. Then, we estimated for each of these models how a voxel's response to the polar angle and eccentricity stimulation should look over time, if it was responsive to that visual portion. In a next step, we computed the correlations between every V1 voxel's actual time course, and the predicted time courses for all portions of the visual field. For every voxel, we then determined which visual portion it was most responsive to, based on which model explained most of the variance of the voxel's responses ($r^2$) over time.

Using this approach, we identified preliminary ROIs by determining those fMRI voxels whose population-receptive fields (with a spread of 1 standard deviation) fell within those portions of the visual field that we were interested in (Fig. 1C). To shield from excessive levels of noise in our models, we thresholded $r^2$, such that only those voxels whose best model's fit exceeded the threshold were included. We chose an $r^2$-value that was as high as possible, while at the same time ensuring that there were still enough voxels above the threshold in each subject. In Experiment 1, this threshold was $r^2 = 0.2$. In Experiment 2, where the visual area of interest was smaller, we had to lower the threshold to 0.1, as there were a number of participants in which no or too few voxels (≤10 across both hemispheres) exceeded the threshold of 0.2, making an estimate of the ROI impossible otherwise. This thresholding procedure is very conservative and protects against noise. However, it also has the consequence that it mostly identifies voxels in the superficial portions of grey matter (see Supplementary Fig. 3 for an illustration of the effect of different $r^2$ thresholds). This is likely because GE-EPI data generally yield stronger responses in the superficial depths compared to voxels in deeper portions[27]. To ensure that all cortical depth layers were represented in the ROI without any biases towards superficial voxels, and to correct for the fragmented nature of voxel identification, we used the following procedure. First, we projected the functional voxels in question onto the inflated cortical surface. Second, we manually drew boundaries around the patches and labelled them. Last, we projected these patches from the two hemispheres back into volume space, where we combined them into one ROI and proceeded with the cortical depth-layer segmentation. It is known that neurons are organised in hypercolumns[56], which means that neurons in the deeper layers represent the same visual portion as those 'stacked' on top of them in the more-superficial layers. As a consequence, it is neuroanatomically justified to include voxels from all layers for a portion on the cortical surface that has been identified as a ROI. Further, the retinotopic structure of the cortex along the cortical surface is known to be smooth and change only gradually[41]; the fragmented nature of the raw ROIs derived from population-receptive field mapping is thus thought to be mostly due to noise. Overlaid retinotopic maps and the ROI's anatomical position on the cortical surface were also used for additional validation of our approach. As another sanity check, we also compared the location of the pRF-defined ROI with the region identified when using a target area vs. surround area contrast (this contrast could be obtained from the chequerboard stimuli shown at the end of each run). Furthermore, we conducted cross-checks using lower thresholds for $r^2$ to confirm that our approach was adequate. As shown in Supplementary Fig. 3, the pRF-defined regions grew into the deeper grey matter portions in question when the $r^2$-threshold was lowered.

In each individual of Experiment 1, we identified 2 ROIs: first, a foveal ROI to examine the information content of the V1 portion that represents the central visual field. This ROI was computed from the same visual portion that was mapped using the target vs. surround contrast from the task-related experimental runs. It spanned a radius of 2.1° from central fixation (2.6° in 4 participants). As a "sanity check", we visually cross-checked the location of the central ROI mapped by pRFs with the one estimated by the target vs. surround contrast.

Second, we determined a peripheral ROI, located at the four illusory boundaries of the illusory square. The visual portions presented four semi-circles, with the straight edge positioned along the illusory boundary (at a distance of 4.2° to the central fixation cross; 4.7° in four participants). Due to the head coil in the fMRI scanner, which limited visibility of the lower (and, sometimes, upper) central part of the screen in most participants, the majority of the identified voxels had their pRF in the right or left semi-circles, and fewer were located in the lower and upper ones.

In Experiment 2, we identified one foveal ROI that represented the semi-circle-shaped visual area along the centrally located illusory contour. Similar to Experiment 1, this ROI was computed from the same visual portion that was mapped using the target vs. surround contrast from the task-related experimental runs. It encompassed the central fixation cross; the straight edge of the semi-circle had a total length of 3.58°, passing the central fixation cross at a distance of 0.68° at its closest point. Again, like in Experiment 1, we visually cross-checked the location of the central ROI mapped by pRFs with the one estimated by the target vs. surround contrast as a "sanity check". To ensure that differences in voxel numbers between layers do not explain our results, we ran a linear mixed effects analysis testing voxel number by depth and ROI (voxel number ~ depth + roi + roi:depth, random factor: -1 + depth + ROI+ depth:ROI | subject). The model confirms that the effect of depth is non-significant (beta = 24.58, 95% CI [-120.84,170.00], $t(225) = 0.33$, $P = 0.739$). There was also no significant interaction between depth and ROI ($P > 0.33$, Supplementary Fig. 16).

## Cortical depth-layer segmentation
After creating a cortical thickness map, BrainVoyager's high-resolution cortex depth grid sampling tool was used to segment the grey matter of the ROIs into 6 equidistant cortical depth layers (at 0.1, 0.26, 0.42, 0.58, 0.74, and 0.9 depth). This was done on the T1-weighted anatomical image, to which the functional images had been aligned. Note that with the coarse resolution of high-resolution MRI compared to histological methods, cortical depth sampling based on equidistant sampling results in only slight differences compared to equivolume sampling[57]. The comparatively coarse resolution also has the consequence that grids at specified depth levels partly use the same voxels as the neighbouring grids, which means that they are not fully independent. For example, some voxels that were included in the grid at 0.9 depth may also have been part of the grid at 0.74 depth.

## Univariate and multivariate analyses of ROI activity
To investigate univariate differences across the 6 cortical depth layers of the ROIs in the different conditions, we computed a GLM analysis, modelling each condition with canonical (i.e., two-gamma) hemodynamic response functions (HRF), time-locked to the presentation of the stimuli in the experimental runs. The default settings of the NeuroElf toolbox were used (time to response peak: 5 s, time to undershoot peak: 15 s, positive-to-negative-ratio: 6, onset delay: 0 s, no derivatives). The two colours in each condition were modelled separately, resulting in a total of 10 factors of interest. The target and surround mapping block at the end of the experimental runs were regressed out as additional factors.

For the multivariate pattern analysis, a linear SVM classification with default parameters was computed (LIBSVM toolbox[58], v2.86). Voxels whose raw mean BOLD signal intensities were below 100 were removed prior to the analysis. Beta weights for every trial were then estimated in a GLM analysis. Before entering the data into the SVM classifier, the beta weights were normalised by rescaling the values between −1 and 1. Classification models for the different depths were trained using the C-SVM method (cost parameter = 1) with a linear kernel, implemented in the LIBSVM toolbox, v2.86[58]. Tolerance for termination was 0.001 (default setting) and cost parameters were equal across classes (i.e., no applied weighting scheme for the different

conditions). Cross-validation was performed in a leave-one-run-out manner. The reported SVM accuracies were averaged across cross-validation folds.

Both the univariate and multivariate analyses were conducted using custom-written scripts in MATLAB R2015b & R2016 in connection with BVQX v0.8b /NeuroElf v0.9c toolboxes.

## Illusory perception vs. mock illusion decoding and visual projections
Previous work has found that illusory contours involve a selective activation of deep V1 layers[7]. In contrast, in our multivariate pattern analysis, we can only decode illusory colour at superficial depths in our study. However, differences in the analyses of these two studies could have contributed to these differences in findings. In our study, we decoded two illusory colours against each other, whereas Kok et al. looked at (univariate) activity differences between the condition when an illusory figure was present and when it was not. For better comparability with Kok et al.'s results, we ran an additional SVM classification analysis, where we decoded the illusory perception condition against the mock version of the illusion. As the stimuli in our study were presented in two different colours, we could compute SVM classification accuracies across the 6 cortical depths twice, once for the red stimuli, and once for the green stimuli. This also allowed us to assess the reliability of the analysis. See Supplementary Fig. 4A, B and Supplementary Note 1 for results. In addition to computing SVM classification accuracies, we also projected voxel influence on SVM classification[59] from the different cortical layers into visual space using a weighted average method based on voxel pRF models[60] (see Supplementary Fig. 4C). Instead of displaying voxel influence in cortical space, this approach thus translates voxel influence into visual space, giving an idea of which visual field portions (or pixels) contribute most to SVM classification. Here, we were interested in which sub-portions of the visual field areas represented in the peripheral V1 ROI were particularly influential for SVM classification. The method takes the following form:

$$y_i = \frac{\Sigma_j (w_{ij} x_j)}{\Sigma_j w_{ij}}$$

where $y_i$ is the activity at a visual field pixel $i$, and $x$ is a vector of SVM weights converted to voxel activity. Note that the term *activity* here does not refer to BOLD activity, but to the influence that different portions of the visual field have on SVM classification analyses. $w$ is a matrix of vectorised pRF functions for the voxels in $x$, with columns $j$ representing voxels, and rows $i$ representing how much a particular visual field pixel is represented by a particular voxel. $y_i$ can become arbitrarily inflated when $\Sigma_j w_{ij}$ is extremely small, as is the case in areas of the visual field without sufficient pRF coverage. We therefore truncated each pRF function to have only weights within $2\sigma$ of its centre and defined $y_i = 0$ for pixels with $\Sigma_j w_{ij} = 0$.

## Statistical testing
Our study aimed to explore the question in which cortical depth layers of our regions-of-interest perceptual/imaginary/illusory stimulus information was decodable. To answer this question, we tested the mean SVM classification accuracy across subjects for each stimulus condition at each cortical depth against chance level (50%), using bootstrapping of the mean with 10,000 samples. Such a non-parametric approach instead of standard parametric tests is recommended for comparing classification accuracies in low sample sizes, as distributions tend to be skewed[61]. The statistical significance level of $\alpha = 0.05$ (one-sided) was corrected for multiple comparisons using false discovery rate (FDR; 60 comparisons across all six layers, five conditions, and two regions of interest in Experiment 1; 30 comparisons across all six layers and five conditions in Experiment 2). As

statistical significance was assessed in a one-sided fashion to compare mean decoding accuracy against chance level in the bootstrapping analysis, we provide the 90% confidence intervals (instead of the 95% confidence intervals) which indicate the 5% and 95% brackets of the distribution of bootstrapped means. Paired $t$ tests were conducted to compare classification accuracies for imagery and illusory perception at deep and superficial depths at which imagery and illusory perception showed significant above-chance decoding ($t$ tests could be used as the classification differences did not violate the normality assumption).

In a second-level analysis, we then fitted a linear mixed model (estimated using REML and nlminb optimizer) to predict decoding accuracy with depth, experiment and stimulus condition using R and the lme function of R's nlme package. This was done to compare the two critical conditions—mental imagery and illusory perception—more directly at the different cortical depths. Computing a linear mixed model has the advantage of avoiding the multiplication of tests and therefore multiple comparisons by using parameterisation across layers and between experiments[62]. This enables us to harvest the power of accumulating evidence. The parametrization across cortical depth was possible because the number of cortical depths we define is arbitrary, these depths are partially overlapping, and they only have a statistical correspondence with histology. The second-level analysis approach also allowed us to examine any statistical differences between the two experiments. To compute the model, we first pooled the data of the critical conditions from the two experiments—i.e., mental imagery decoding in the central ROI and illusory perception decoding in the peripheral ROI from experiment 1, and mental imagery and illusory perception decoding in the central ROI from experiment 2. We included as predictors an intercept, depth, experiment, stimulus condition, the interaction between experiment and depth, between stimulus condition and depth, between experiment, stimulus condition and depth. As random effects across participants, we included intercept, depth and the interaction between stimulus condition and depth.

Further, using R with rstatix, 2 (foveal vs peripheral ROI) x 10 (stimulus condition in each of the two colours) x 6 (cortical depth) repeated measures ANOVA was used to examine the results of the univariate analysis (with 10 stimulus conditions instead of 5 to distinguish between red and green colour; Supplementary Fig. 6). Prior to computing the repeated measures ANOVA, we used bestNormalize[59] to transform the data to obtain normal distribution of the response variable. Post-hoc pairwise $t$ tests to directly compare illusory perception condition against the mock and amodal condition at different depths were corrected for multiple comparisons using the false discovery rate (FDR).

To analyse the relationship between imagery strength and subjective vividness in the behavioural pre-test data, we used Spearman rank correlation for the between-subject analysis (Supplementary Fig. 1C), as the normality assumption for the imagery strength values was violated in Experiment 1 (Shapiro–Wilk normality test: $W = 0.90$, $P < 0.001$). Further, to examine the trial-by-trial relationship between imagery-induced priming and vividness, we computed a linear mixed effects model with a 2 (group) × 4 (vividness rating) design (Supplementary Fig. 1D). As a random effect, intercepts for subjects were modelled. We inspected the plots of the residuals visually and computed Shapiro–Wilk tests to ensure they did not show deviations from homoscedasticity and normality. As the residuals were not normally distributed (Experiment 1: $W = 0.956$, $P < 0.001$, Experiment 2: $W = 0.961$, $P < 0.001$), an ordered quantile normalisation was applied to transform the data before computing the model using bestNormalize[63]. The analysis of the pre-sample data of Experiment 2 was conducted analogously.

## Reporting summary

Further information on research design is available in the Nature Portfolio Reporting Summary linked to this article.

## Data availability

The data generated in this study have been deposited in the EBRAINS Knowledge Graph, https://kg.ebrains.eu/search/instances/Dataset/de7a6c44-8167-44a8-9cf4-435a3dab61ed. Access to this resource requires free user registration at https://www.ebrains.eu/page/sign-up. The source data presented in the figures are provided with this paper and is available at https://gitlab.com/joebee/7t-imagery-illusory-in-v1-layers. Source data are provided with this paper.

## Code availability

The code of this study is available at https://gitlab.com/joebee/7t-imagery-illusory-in-v1-layers.

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

## Acknowledgements

This project has received funding from the European Union's Horizon 2020 Framework Programme for Research and Innovation under the Specific Grant Agreement No. 720270 (Human Brain Project SGA1): awarded to L.M., 785907 (Human Brain Project SGA2): awarded to L.M., and 945539 (Human Brain Project SGA3): awarded to L.M, and Biotechnology and Biological Sciences Research Council (BBSRC BBN010956/1) 'Layer-specific cortical feedback' awarded to LM (with LSP). We thank Anna Makova for her assistance in the data acquisition for

the second fMRI experiment. We thank Jane Alfred for helpful comments on the manuscript.

## Author contributions

J.B. and L.M. independently conceived the study, and then designed the study together. L.S.P. conceived the amodal version of the neon colour-spreading stimulus. J.B. acquired the pre-test behavioural data and fMRI data of the first experiment; M.S.L. acquired the pre-test behavioural data of the second experiment, and J.B., A.T.M., and M.S.L. acquired the fMRI data of the second experiment. J.B. pre-processed and analysed the data of both experiments. A.T.M. computed the visual projections, and C.A. contributed the second-level linear mixed effects model across the two experiments. J.B. wrote the manuscript, and all authors edited the manuscript.

## Competing interests

The authors declare no competing interests.
