## [Peer Review File · Nature Communications]

Cortical depth profiles in primary visual cortex for illusory and imaginary experiencesReviewer #1 (Remarks to the Author):

The study by Bergmann et al. examines the patterns of activity at different depth of V1 in relation to perception, illusory perception and mental imagery. Participants were asked to view the real coloured discs, to view the neon colour spreading illusion, or to imagine coloured discs, whilst their visual cortex was imaged using 7T fMRI. The researchers found above chance decoding in the deep layers of V1 for imagery content, and above-chance decoding in superficial layers for illusory content. These findings should demonstrate that different types of subjective experience can be dissociated across different cortical layers.

The claims made in the paper are interesting, and the approach of trying to dissociate the processing associated with different sources of feedback across the cortical depth using 'laminar' fMRI is quite novel (though see <https://www.biorxiv.org/content/10.1101/2022.04.13.488155v1> for work along similar lines). The manuscript is excellently written and provides sufficient details for understanding and judging the methodological soundness. My major concern is related to the way statistical inference is performed to support the main claims of the study, as detailed below. I also raise a few other major and minor points that should be addressed.

Major

1. Statistical analysis. Currently the major claims of the paper are mainly based on either observing or not observing significant above-chance decoding accuracy at a specific depth. This, in my opinion, is not sufficient to claim depth specificity. Statistical significance is merely a probability, and we can't claim the presence or absence of an effect just because it does or does not pass an (arbitrary) significance threshold we set. Based on the current logic, we would have also had to conclude that amodal condition is decodable in the most superficial layer, while illusion condition in the second most superficial layer (Figure 2C), which is a very unlikely scenario given the coarse imaging resolution and signal mixing in the adjacent depths. A proper analysis in my opinion would be an interaction in a repeated measures ANOVA with depth and task as factors for experiment 2, with pairwise post-hoc comparisons of conditions at every depth, similar to what is done for beta estimates (univariate analysis) in supplementary figure 6. Unfortunately, the authors present such an ANOVA result only for experiment 1 (lines 181-182), which is irrelevant for the reasons explained in my next point. I could not find the results of these analyses anywhere in the manuscript for Experiment 2. It would also be nice to have the full statistical information on the ANOVA (main effects and interaction).

2. Effects within the same ROI. As the authors acknowledge themselves, it is critical to show the dissociation of effects across depths within the same ROI. Otherwise, any differences between ROIs can explain the differences in decoding accuracy across layers. For example, there may be a different number of voxels per layer in each ROI, or ROI alignment with anatomical depth definition can vary. The readers need to be aware of the fact that the actual main result is Figure 1C, where the region of interest is identical for both tasks.

3. On a related note, I could not find any information on the number of voxels per ROI and per layer of each ROI, only a statement that some voxels were excluded if they had intensity below 100 (what was actually the motivation for voxel exclusion?). Can the authors ensure that the SVM had a similar number of voxels per layer to work with, and if not, can they show that the number of voxels can't explain the observed results?

Minor

1. L1030-1033, Supplementary figure 5: The authors' explanation of the lack of decodability for imagery condition when trained on perception is not consistent with the literature. Imagery decoding has been shown in laminar fMRI context before (Iamshchinina et al., 2021, <https://www.nature.com/articles/s42003-021-02582-4>), so the authors' explanation that 3T studies can show this effect because of their coarse resolution, while the current study can't because of the high resolution, does not apply. The study by Iamshchinina et al., 2021 reports a medium effect size, which suggests that the current study may be underpowered in this regard. Can the authors rule out or acknowledge this explanation?

2. L647 onwards, Supplementary figure 4: I think there are multiple reasons for the discrepancy between the results of Kok et al. 2016 and the current study in terms of cortical depth representing illusory perception, which remain unaddressed. Another possibility is that the current experimental design focuses on the colour information, which is available primarily in the upper

layers of V1 (due to the location of the colour blobs), while Kok et al.'s design takes advantage (presumably) of the orientation selectivity, which is distributed across the whole orientation column. See e.g. Olman et al., 2012 (<https://doi.org/10.1371/journal.pone.0032536>) for context. In any case, I think that if the authors want to compare the effects with those reported by Kok et al., they should perform a univariate analysis comparing illusory perception with amodal/mock condition, instead of MVPA.

3. Supplementary figures 7 and 8 with individual subject plots suggest that there are differences across ROIs in their position with respect to the vasculature (e.g. subject 9 shows a decrease towards superficial layers for the centre, and a peak in the middle for the periphery)

4. L587-589 and L596-597: how were potential mismatches between the pRF-derived and checkerboard localizer-derived ROIs handled?

5. Supplementary Fig. 4C: Perhaps the authors can provide more details on why the t-values are plotted. I assume these are the t-test results? What was tested against what? I was expecting to see something like classifier weights.

Formatting/typos/suggestions:

1. Supplementary figure 14 caption "Padj" lacks a subscript formatting

2. Sometimes a dot after "Fig" is missing

3. There are switches between the British English and American English spelling – it is better to stick with one (examples: coloured, familiarize, behavioural)

4. Asterisks in all figures that indicates significant effects could be a bit bigger

5. L238: "formation of a more malleable 'inner' sensory world that we can keep separate to our ongoing" -> separate from

6. L445: "illusory and control conditions would not fall within the receptive fields of the foveal V1 region" - > delete "not"?

7. L543: retinopic -> retinotopic

8. L666: "not in line Kok's et al.'s " - > not in line with Kok et al.'s

9. L673: do you mean Fig. 4C?

10. L702: "above-decoding " -> above-chance decoding

Reviewer #2 (Remarks to the Author):

This is a very interesting manuscript where the authors hypothesized that information about non-physical visual experiences, like imagery and illusions, should be distinctly dominant in the layers that receive feedback from other brain areas. Specifically they find that imagery content is decodable mainly from deep layers of V1, and if illusory content is decodable mainly from superficial layers. Their data, which is very well substantiated by additional evidence shown as supplementary material are consistent with the notion that feedback signals in superficial and deep layers may serve distinct roles in information processing. This study is methodologically very sound and carefully conducted.

I have only a few concerns:

1. This study is in stark contradiction with Kok et al. who showed activity differences between the condition when an illusory figure was present and when it was not, and found differential activation of deep V1 layers. The authors believe that differences in the analyses between the two studies have driven these differences, and although they ran additional analyses to show that superficial layers are more likely to encode illusory contours, the differences between the two studies remain unclear. They should further discuss why analyses issues would lead to such opposite results and whether experimental design issues might also help explain differences.

2. The study design is mainly data driven, and the fact that representations can be decoded using data driven approaches leaves open the possibility that encoding may be performed upstream, as actually also pointed out by the authors. Would it be possible for the authors to show decoding from such upstream regions? The authors should address the issue whether these regions can be identified using their data.

3. The sample size is not very large and effect sizes are moderate. The authors should provide a justification for sample size and an explicit discussion of power and effect sizes.
4. In the 'amodal' version of the stimulus the white contour in the area between the pacman-like rings does not fully break the illusory experience but instead generates the perception of occlusion. This is partly acknowledged but should be further discussed also in terms of the observed experimental results.
5. Regarding the 3 main conditions (imagery, perception, and illusory perception) what would the authors expect if participants would imagine an illusory contour? Although this is not a required experiment they should discuss this in the context of future work.
6. The "imagery" profile in plot in Figure 2C seems relatively flat. What could be the reason for that?

Reviewer #3 (Remarks to the Author):

Bergmann et al. performed sub-mm resolution fMRI in humans during mental imagery, when presented with visual illusions, or when real stimuli were projected on the retina. In striate cortex, they observed differences in activation patterns as a function of cortical depth between these three conditions. Mental imagery can be decoded from fMRI activity in superficial V1 layers, while activity in deeper layers can be used to decode visual illusions. The authors suggest that different microcircuits are recruited during subjective experiences evoked by imagery and stimuli inducing illusions. While the results obtained with the illusory objects confirmed previous research from the same group, it contradicted the results from the group of Floris de Lange.

In general, the study has been executed with great care. The behavioral assessment for the strength of imagination of individual subjects is highly commendable. Unfortunately, this information is not used to explain the inter-individual variability of the fMRI results. The analyses are state of the art. Moreover, it is also commendable that most data of individual subjects are shown (but see some suggestions below).

Although the study is nicely and carefully executed and although I tend to believe the results, I suspect that the 'real' functional effects are exceedingly subtle and very hard to capture with fMRI, even using high field scanners. My main reason for my reduced enthusiasm is the high inter-subject variability as shown in the supplementary information. The illusory and mental imagery data are quite variable across subjects, sometimes even showing opposite results or trends. One has to consider that the 'average' group result, on which the main conclusions are based, are simply a reflection of the relatively small group of subjects being tested. This may also explain the differences between the present study and that of de Lange's group. In other words, one should consider the possibility that really large numbers of subjects need to be tested before conclusive statements can be made. Hence the study may be under-powered. A stringent power analysis is warranted in this regard. Also, a comparison of the results of both hemispheres separately for each subject might provide insightful information. There may be considerable inter-subject variability, certainly concerning the strategies to perform a mental imagery task and the relevant circuitry being employed. However, one does not expect major differences across hemispheres of the same subject. Inter-hemispheric correlations should be higher than inter-subject correlations in decoding strength as a function of laminar depth. The major question is whether 16 subjects is enough to reveal conclusive results.

Another major issue I have is that, by definition, the displays during the different conditions was quite different. How can one exclude the possibility that results are explained by these different stimulation conditions. The subjects make eye-movements - which were difficult to measure and not considered. Even small differences in retinal stimulation (in the fovea) will evoke different activity levels independent of the illusory or imagery conditions. Moreover, influences from outside the classical receptive field of the foveal V1 neurons, induced by the differences in the display, might modulate activity independent of the subjective experiences during the different conditions. To really exclude the influence of saccades and extra-classical receptive field phenomena, the

stimuli should be much larger. The question is whether this is feasible.

Along the same lines, why is a faint grey contour shown during the mental imagery condition? A 3.5 deg radius circle may be large enough to evoke activity in foveal representations, certainly when subjects make eye movements.

Is an ISI of 7-8 seconds enough to clear after-images following the 'real' perceptual and the illusory contour conditions?

The behavioral pre-test for mental imagery is commendable. However, is there a correlation between the score and the fMRI data? 32 hemispheres should be sufficient to get a reliable estimate.

Did the authors consider a decoding analysis as a function of eccentricity & laminar depth?

Line 385: Fig 2A \diamond 1A

Line 577: It is unclear why the analyses jump from 3D \diamond 2D \diamond 3D. Staying within 3D should reduce errors, no?

I'm somewhat puzzled by the segregation between some 'parts' of the same ROI (e.g. S11 in Fig 3). One would expect a continuous uninterrupted ROI - unless it is an illusion induced by the 2D section. It would be highly informative if the ROIs for all subjects are shown in this format and also in flattened 2D format.

Line 650: A potential explanation for the differences between the current study and Kok et al., is that the latter looked at univariate activity differences evoked by an illusory contour versus nothing (while in the present study SVMs were used to decode two illusory contours from each other). However, to resolve the issue, the authors could also perform the same univariate analyses as Kok et al.

REVIEWER COMMENTS

Reviewer #1 (Remarks to the Author):

The study by Bergmann et al. examines the patterns of activity at different depth of V1 in relation to perception, illusory perception and mental imagery. Participants were asked to view the real coloured discs, to view the neon colour spreading illusion, or to imagine coloured discs, whilst their visual cortex was imaged using 7T fMRI. The researchers found above chance decoding in the deep layers of V1 for imagery content, and above-chance decoding in superficial layers for illusory content. These findings should demonstrate that different types of subjective experience can be dissociated across different cortical layers.

The claims made in the paper are interesting, and the approach of trying to dissociate the processing associated with different sources of feedback across the cortical depth using 'laminar' fMRI is quite novel (though see <https://www.biorxiv.org/content/10.1101/2022.04.13.488155v1> for work along similar lines). The manuscript is excellently written and provides sufficient details for understanding and judging the methodological soundness. My major concern is related to the way statistical inference is performed to support the main claims of the study, as detailed below. I also raise a few other major and minor points that should be addressed.

We thank the reviewer for their careful consideration and encouraging feedback. We have responded to all points below. Importantly, with a new analysis, we have addressed R1's concern related to the statistical support of our main claims.

Major

1. Statistical analysis. Currently the major claims of the paper are mainly based on either observing or not observing significant above-chance decoding accuracy at a specific depth. This, in my opinion, is not sufficient to claim depth specificity. Statistical significance is merely a probability, and we can't claim the presence or absence of an effect just because it does or does not pass an (arbitrary) significance threshold we set. Based on the current logic, we would have also had to conclude that amodal condition is decodable in the most superficial layer, while illusion condition in the second most superficial layer (Figure 2C), which is a very unlikely scenario given the coarse imaging resolution and signal mixing in the adjacent depths. A proper analysis in my opinion would be an interaction in a repeated measures ANOVA with depth and task as factors for experiment 2, with pairwise post-hoc comparisons of conditions at every depth, similar to what is done for beta estimates (univariate analysis) in supplementary figure 6. Unfortunately, the authors present such an ANOVA result only for experiment 1 (lines 181-182), which is irrelevant for the reasons explained in my next point. I could not find the results of these analyses anywhere in the manuscript for Experiment 2. It would also be nice to have the full statistical information on the ANOVA (main effects and interaction).

You are raising very important points and we will reply to them extensively:

(1) Multiple comparisons and supersampling of layers

We are facing the problem that we want to sample a sufficient number of layers to be sensitive to gradual changes in the underlying neurophysiology. By increasing the number of layers we are increasing the number of statistical tests and therefore alpha inflation. Our layers of cortex are not independent but partially overlapping, but the number of samples is arbitrary; some use 3 bins, others use 10 or more. A fair comparison for multiple comparisons becomes essential. For example, an overly conservative Bonferroni correction would punish high sampling of layers, but selecting a low number of layers would be insensitive and blur the underlying neurophysiological effects. We therefore suggest a model testing approach (below).

(2) Significant classification in some layers vs significant testing of profiles

What can be learned from significant decoding in a layer; is it meaningful or random that we find patterns of brain activity related to specific stimulus encoding in particular layers? Firstly, we have a set of predictions, based on our own previous results and those of others. We designed the experiment to arbitrate between our results and those of the lab of Floris deLange (more below). We have a solid conceptual expectation in terms of microcircuit function which initiated computational and multispecies neuroscience and led to replication across species (replication of our layer specific amodal completion results in monkeys V1 [<https://www.biorxiv.org/content/10.1101/2022.11.21.517305v1>] and mice V1 [in preparation] using 2-photon calcium imaging, and also related mice visual cortex data that feedback is functionally relevant and targets superficial layers for figure ground segregation (Kirchberger, Roelfsema, 2021). Moreover, we replicated our main findings across two experiments (and they are a replication in part to previous findings in humans from our lab but with different stimuli, Muckli et al., 2015).

But how sensitive or misleading are our statistical tests if they are not protected against alpha level inflation (see above) by use of one single unified ANOVA? Instead of significance testing across multiple comparisons a better approach might be to test a model and accumulate evidence across subjects and layer-sampling. The reviewers' suggestions have motivated us to provide such a model test. This statistical point speaks to a wider scientific discourse, unspecific to our paper. Within the hypothesis testing framework that is still dominating, our current approach of significance testing has merit. However, we appreciate the reviewer's point, and as such, we have augmented our previous statistical analysis to inform on the relative levels of evidence for our effects (next point 3). One solution to the problem of multiple comparisons, and indeed to harvest the power of accumulating evidence, is to parametrize, which we do here across layers and between experiments. This new analysis has strengthened support for our main claims, and we have added it to the manuscript in a new figure 2D.

(3) Model testing and evidence accumulation:

Because the number of cortical depths we define is arbitrary, these depths are partially overlapping and they only have a statistical correspondence with histology, we replaced the previous depth-wise ANOVA approach with a mixed effect linear model parameterized across cortical depth. With this approach we avoid the multiplication of parameters at each depth level, resolving many multiple comparison issues, and we accumulate evidence from both experiments.

The first level bootstrapping analysis showed us that, in the critical experimental conditions (imagery and illusory perception), and with respect to our control conditions, illusory content was significantly decoded only in the superficial cortex while imagery content was significantly decoded only in the deeper layers of cortex. This was replicated within both experiments. Now, our second level model allows us to investigate whether there is a significant interaction between decoding accuracy across cortical depth and these two conditions while controlling for the effect of the experiment as well as random effects over participants.

After fitting this model we see a significant interaction between condition and depth ($t(283) = 3.70$, $p < .001$). In other words, we find that the slope significantly differs between the mental imagery and illusory contour conditions (figure below **Model predictions**). Note that this is true while controlling for any effect of experiment 1 vs experiment 2, as well as random effects across participants. In fact there is no significant main effect of experiment, no significant interaction between experiment and depth, or between experiment, condition and depth. Importantly, experiment therefore doesn't significantly affect slope. There is a significant effect of condition, and a significant interaction effect between experiment and condition (more accuracy for mental imagery in the second experiment), all of which is orthogonal to the slope effect we are interested in.

In conclusion, we find with this model an interaction effect in which the slope (i.e. difference in decoding accuracy across cortical depth) significantly differs between the critical experimental

conditions (mental imagery and illusory perception). This analysis complements the first level bootstrapping analysis showing that illusory content is significantly decoded only in the superficial cortex while imagery content is significantly decoded only in deep cortex. We have added this text and figure to the revised manuscript.

Figure 2D: Model predictions. We fitted a linear mixed model (estimated using REML and nlminb optimizer) to predict decoding accuracy with depth, experiment and stimulus condition (mental imagery and illusory perception). Model predictions for both conditions (across experiments) are plotted alongside the experimental data.

2. Effects within the same ROI. As the authors acknowledge themselves, it is critical to show the dissociation of effects across depths within the same ROI. Otherwise, any differences between ROIs can explain the differences in decoding accuracy across layers. For example, there may be a different number of voxels per layer in each ROI, or ROI alignment with anatomical depth definition can vary. The readers need to be aware of the fact that the actual main result is Figure 1C, where the region of interest is identical for both tasks.

We agree with the reviewer that our evidence from the same region of interest is important for our overall story. We highlight in the manuscript text that, even given the significant results of experiment 1 in different ROIs, a more stringent test would be to examine decoding of illusory and imaginary feedback signals in the same region of V1. That said, in our new linear mixed effects model analysis, we do not see that our findings differ between experiments. In our new analysis mentioned in Point 1 above, we find no significant main effect of experiment ($t(283) = -1.35$, $p = 0.178$), no significant interaction between experiment and depth ($t(283) = 1.47$, $p = 0.143$), or between experiment, condition and depth ($t(283) = -1.43$, $p = 0.155$), (fig 2 below **Model predictions for individual experiments**). We also now confirm that voxels numbers do not differ between layers (see response to point 3 next).

Supplementary Figure 13. Model predictions for individual experiments when a linear mixed model was trained to predict decoding accuracy with depth and stimulus condition for individual experiments 1 and 2 (exp 1 & exp 2), where the regions of interest differed. Model predictions are plotted alongside the experimental data. Mental imagery data and model predictions for both experiments are shown in pink (left), and illusory perception data and model predictions are shown in blue (right).

3. On a related note, I could not find any information on the number of voxels per ROI and per layer of each ROI, only a statement that some voxels were excluded if they had intensity below 100 (what was actually the motivation for voxel exclusion?). Can the authors ensure that the SVM had a similar number of voxels per layer to work with, and if not, can they show that the number of voxels can't explain the observed results?

The reviewer asks if our results could be accounted for by differences in voxels numbers between layers or ROIs. We have conducted a new analysis towards this question. A linear mixed model analysis testing voxel number by depth and ROI confirms that the effect of depth is non-significant (beta = 24.58, 95% CI [-120.84, 170.00], $t(225) = 0.33$, $p = 0.739$; Std. beta = 0.04, 95% CI [-0.08, 0.16]), (fig 3). The same holds for the interaction between depth and ROI ($p > .33$).

New Supplementary Figure 16. Number of voxels across cortical depths in experiment 1 and 2. The blue circles represent the voxel numbers per depth of the individual subjects. The pink lines/circles show the group means across subjects, with error bars denoting \pm SEM. Deeper cortical depths are shown towards the left of each plot. There was no significant main effect of depth, indicating that the number of voxels was not significantly different between cortical depth layers ($p > 0.73$). There was also no significant interaction between ROI and depth ($p > .33$).

We have added to the manuscript text (Methods):

“To ensure that differences in voxel numbers between layers do not explain our results, we ran a linear mixed model analysis testing voxel number by depth and ROI (voxel number \sim depth + roi + roi:depth, random factor: \sim 1 + depth + ROI+ depth:ROI | subject) . The model confirms that the effect of depth is non-significant (beta = 24.58, 95% CI [-120.84, 170.00], $t(225) = 0.33$, $p = 0.739$; Std. beta = 0.04, 95% CI [-0.08, 0.16]). There was also no significant interaction between depth and ROI ($p > .33$).”.

Minor

1. L1030-1033, Supplementary figure 5: The authors’ explanation of the lack of decodability for imagery condition when trained on perception is not consistent with the literature. Imagery decoding has been shown in laminar fMRI context before (Iamshchinina et al., 2021, <https://www.nature.com/articles/s42003-021-02582-4>), so the authors’ explanation that 3T studies can show this effect because of their coarse resolution, while the current study can’t because of the high resolution, does not apply. The study by Iamshchinina et al., 2021 reports a medium effect size, which suggests that the current study may be underpowered in this regard. Can the authors rule out or acknowledge this explanation?

Whilst the added layer-specificity of our data has important conceptual and technical relevance for comparison with the previous studies that we currently mention, R1 adds another layer fMRI reference speaking to this point. The suggested paper used 3 bins to approximate layers, whilst we used 6, and so they likely still have more voxels that contribute signal to a layer, with their layers still pooling signals from voxels across larger regions than in our data. Furthermore, there are important stimulus differences to consider; (1) the imagery process for static imagery (that we use) might be different to the mental rotation task, evidence in support of this comes from congenital aphantasia where participants who use analogical strategies perform with similar accuracy as controls in mental rotation tasks (Pounder et al., 2022). (2) As the reviewer mentions in the next comment, there may be some differences in the neurons responding to colour stimuli versus to grating stimuli, although it is not clear how far that would show in layer-specific processing differences (see our response to the reviewer’s next comment).

Nevertheless, we agree that by approximating cortical compartments that sample feedforward and feedback processing separately (as we do), it is relevant to include this paper. As such, we have edited the manuscript as follows:

“For mental imagery, cross-classification did not reach significance, neither for illusory perception, nor for perception ($p_{adj} < .05$). This is noteworthy, as such significant cross-classification effects have been reported in previous studies, which used larger voxels and did not distinguish between different cortical depths^{29,64}. When using smaller voxels and decoding for each layer separately, voxels are less prone to pooling signals over a larger space and from a larger range of cortical depths. In this case, it may be that the (layer-wise) divergences between these two experiences become more apparent. However, with layer resolution data, Iamshchinina et al.⁶⁵ find that mentally rotated gratings are represented mostly in outer cortical depths (i.e. superficial and deep), and perceived contents are represented most strongly in the middle cortical depths.”

Pounder, Z., Jacob, J., Evans, S., Loveday, C., Eardley, A. F., & Silvanto, J. (2022). Only minimal differences between individuals with congenital aphantasia and those with typical imagery on neuropsychological tasks that involve imagery. *Cortex*, 148, 180-192.

2. L647 onwards, Supplementary figure 4: I think there are multiple reasons for the discrepancy between the results of Kok et al. 2016 and the current study in terms of cortical depth representing illusory perception, which remain unaddressed. Another possibility is that the current experimental design focuses on the colour information, which is available primarily in the upper layers of V1 (due to the location of the colour blobs), while Kok et al.'s design takes advantage (presumably) of the orientation selectivity, which is distributed across the whole orientation column. See e.g. Olman et al., 2012 (<https://doi.org/10.1371/journal.pone.0032536>) for context. In any case, I think that if the authors want to compare the effects with those reported by Kok et al., they should perform a univariate analysis comparing illusory perception with amodal/mock condition, instead of MVPA.

We agree that there are a number of outstanding questions as to why we do not replicate Kok et al (see more below). Indeed, there are challenges with interpreting univariate analyses in layers, given the robust bias of BOLD signal to the superficial layers due to the vascular structure, hence multivariate approaches are helpful. We also take different approaches to functional alignment and grey/white matter segmentation to those taken by Kok et al.

The reviewer asks us to compare illusory perception with the amodal or mock condition, because Kok et al. found that illusory contours activated the deep layers of V1 in a univariate analysis. We have included such a univariate analysis of BOLD activity levels (Supplementary figure 6), and have now added direct post-hoc comparisons of the illusory condition with the amodal and mock condition at each depth. As we used red and green, our design has the advantage that we can analyse univariate results for the two colours separately, ensuring the robustness of our results. As can be seen in Suppl. Fig. 6, there are significant main effects of depth and stimulus condition and a significant ordinal interaction between the two. The effects are fairly straightforward - perception has the strongest activation, the mock condition has the lowest activation; activation levels increase from deep to superficial layers, and this is more so for those conditions that have stronger overall activation levels.

To more specifically address the reviewer's concern, we also ran post-hoc tests (multiple-comparison corrected) to directly compare activation levels during illusory perception against amodal and mock, separately for the two colours. When comparing illusory perception against the mock condition (peripheral ROI in experiment 1), we found significant differences across all layers in both the red and green version of the stimuli (all $p_{adj} < .001$). When comparing illusory perception against the amodal condition, we again found significant differences across all layers for the green version of the stimuli (all $p_{adj} < .02$); for the red version, only the more superficial layers were significantly different in activation levels (depths 0.1, 0.26, 0.42; all $p_{adj} < .025$). In the second experiment, there were no significant differences between the illusory perception conditions and either of the two control conditions (all $p_{adj} > .1$). We have now added these results to the caption of Supplementary Fig. 6.

We have great respect for the work of Kok et al. and we thought this experiment might be well designed to compare the univariate results of Kok and DeLange and the multivariate results of our lab in conditions that compare amodal and modal completion as well as imagery in one comparable experiment. A replication of the univariate analysis in conditions of modal completion would have had the chance to stand next to the multivariate analysis and we could have had a unifying paper explaining previous results of both labs. However, we were not able to replicate the univariate results of Kok et al. We did, however, find consistent multivariate decoding results across the two experiments for illusory perception against its mock version, see Suppl figure 4. These results were similar in both experiments and in both colour versions of the stimuli, and consistently suggest that superficial layers - *not* the deep layers - seem to be the stronghold here.

Taken together, our univariate and decoding analyses to more directly compare the illusory perception condition against the amodal or mock condition consistently do not support the findings by Kok et al. Rather, our results suggest an involvement of superficial layers in illusory perception.

Lastly, the reviewer raises an important point regarding the difference in stimuli - colour vs. orientation. It is possible that this may add to the disparity in results between Kok et al. and our study. However, please note that while colour blobs may be located in the upper layers of V1, feedforward and feedback projections to and from the parvocellular layers of the LGN - which carry signals for red-green colour vision - are in fact located in the mid and deep layers of V1 (Briggs & Usrey, 2011, *Journal of Physiol*, 10.1113/jphysiol.2010.193599). See below a screenshot from the Briggs & Usrey article which shows the laminar organisation of said projections. Hence, red-green colour signal processing is actually not limited to the upper layers, so it remains an open question as to how the difference in stimuli impacted the divergence in results between Kok et al., and our study. In addition, a focus of colour processing on upper layers of V1 would not explain our finding that imagined colour information is present in deep layers (whereas illusory colour information is present in superficial layers). Also, please note that our findings of a superficial layer involvement in (illusory) boundary processing is also supported by findings in non-human primates and mice that edges between figure and ground lead to additional V1 activity caused by synaptic input into layer 2 & 3 (Self et al., 2013, *Curr Biol*; Schnabel et al., 2018, *Scientific Reports*).

Summing up, while we cannot exclude the fact that a difference in stimulus type may have influenced the different results in some way, we do not believe that it can explain the divergence in full.

Figure 2. Anatomy of feedforward and feedback connections between the LGN and visual cortex (V1)
 A, Nissl-stained section of V1 from the macaque monkey. Corticogeniculate neurones are located exclusively in layer 6. (w.m., white matter.) B, organization of connections between the LGN and V1. The magno-cellular layers of the LGN (1 and 2) are shown in grey, the parvocellular layers (3, 4, 5 and 6) are shown in green and red, the koniocellular layers are located below each of the magno-cellular and parvocellular layers and are shown in blue. Magno-cellular LGN axons (M) terminate in layers 4C α and lower layer 6, parvocellular LGN axons (P) terminate in layers 4C β and upper layer 6, koniocellular LGN axons (K) terminate in layer 4A, the cytochrome oxidase rich blobs and layer 1. The intrinsic connections in V1 maintain the magno- and parvocellular divisions of layers 4C and 6. Neurons in layer 6 of cortex provide feedback to the LGN. Neurons in the upper third of layer 6 project exclusively to the parvocellular LGN layers and perhaps the koniocellular layers. Neurons in the lower third of layer 6 project primarily to the magno-cellular layers and perhaps the koniocellular layers.

(Screenshot from Briggs & Usrey, 2011, *Journal of Physiol*, 10.1113/jphysiol.2010.193599)

We have edited the manuscript as follows:

“Although this superficial layer effect for illusory perception contradicts earlier laminar fMRI findings⁷, it aligns well with results on edge perception between figure and ground from electrophysiology research in mice²⁰ and primates²¹. It should be noted that we use colour stimuli, whereas the contradicting earlier laminar fMRI findings used orientation stimuli. How and whether this may have influenced the results differently remains an open question. Neurons responding to orientation may be distributed throughout the column. However, colour processing, too, appears to involve many layers, with colour blobs being present in upper layers, while feedforward and feedback signals that carry red-green colour information are also processed in mid and deep layers (Briggs & Usrey, 2011).”

3. Supplementary figures 7 and 8 with individual subject plots suggest that there are differences across ROIs in their position with respect to the vasculature (e.g. subject 9 shows a decrease towards superficial layers for the centre, and a peak in the middle for the periphery).

It is possible in experiment 1 that central and peripheral ROIs are more or less close to large veins in subject 9, and deoxygenated blood filters perpendicularly to cortical layers, back towards the pial surface where there are large draining veins. However, we were careful to exclude voxels that were in direct proximity of large draining veins (using a manual segmentation from what could be seen on the T1-weighted images). We are concerned about this but believe that it has been addressed, especially due to the results of experiment 2 where we decode imagery and illusory perception in the same ROI. Furthermore, (1) our retinotopic mapping procedures help to eliminate non-specific signals from draining veins, (2) we do not anticipate that signals from larger blood vessels would induce a particular confounding stimulus-specificity and (3) we have confirmed previously (and in the current manuscript) that difference in BOLD signals dependent on the cortical depths are not predictive of the MVPA decoding patterns in the corresponding depths. (Previously, we also tested this using two different sequences, including a GRASE-3D sequence which has equal signal amplitude across depth layers and improved specificity compared to GE-EPI Muckli et al., 2015).

4. L587-589 and L596-597: how were potential mismatches between the pRF-derived and checkerboard localizer-derived ROIs handled?

We did not observe mismatches between the retinotopic mapping. We found that our pRF-derived ROIs were much more conservative, i.e. the areas were smaller, fitting within the checkerboard localised ROIs.

5. Supplementary Fig. 4C: Perhaps the authors can provide more details on why the t-values are plotted. I assume these are the t-test results? What was tested against what? I was expecting to see something like classifier weights.

Yes, in figure 4C we plot the colour-encoded t-values indicating each visual field *pixel's* influence on SVM classification (across subjects). Visual projections are a transformation of classifier weights of all voxels into visual space using a weighted average model, and are hence a transformation of *voxel* influence to *pixel* influence using each voxel's pRF model (these pRF models are Gaussian). The pRF models of the different voxels (often) overlap strongly, and hence a t-statistics is used for the weighted average model to assess each pixel's influence.

Formatting/typos/suggestions:

1. Supplementary figure 14 caption “Padj” lacks a subscript formatting
2. Sometimes a dot after “Fig” is missing

3. There are switches between the British English and American English spelling – it is better to stick with one (examples: coloured, familiarize, behavioural)
4. Asterisks in all figures that indicates significant effects could be a bit bigger
5. L238: “formation of a more malleable ‘inner’ sensory world that we can keep separate to our ongoing” -> separate from
6. L445: “illusory and control conditions would not fall within the receptive fields of the foveal V1 region” -> delete “not”?
7. L543: retinopic -> retinotopic
8. L666: “not in line Kok’s et al.’s ” -> not in line with Kok et al.’s
9. L673: do you mean Fig. 4C?
10. L702: “above-decoding ” -> above-chance decoding

Thank you for these suggestions, we have made all of these amendments.

Reviewer #2 (Remarks to the Author):

This is a very interesting manuscript where the authors hypothesized that information about non-physical visual experiences, like imagery and illusions, should be distinctly dominant in the layers that receive feedback from other brain areas. Specifically they find that imagery content is decodable mainly from deep layers of V1, and if illusory content is decodable mainly from superficial layers. Their data, which is very well substantiated by additional evidence shown as supplementary material are consistent with the notion that feedback signals in superficial and deep layers may serve distinct roles in information processing. This study is methodologically very sound and carefully conducted.

We thank the reviewer for their kind words and encouraging feedback.

I have only a few concerns:

1. This study is in stark contradiction with Kok et al. who showed activity differences between the condition when an illusory figure was present and when it is was not, and found differential activation of deep V1 layers. The authors believe that differences in the analyses between the two studies have driven these differences, and although they ran additional analyses to show that superficial layers are more likely to encode illusory contours, the differences between the two studies remain unclear. They should further discuss why analyses issues would lead to such opposite results and whether experimental design issues might also help explain differences.

In response to the reviewer, we have edited the manuscript text to further discuss the differences in the results (see more below).

To expand, we have had personal communications with the authors of the study in question, we admire and appreciate their work greatly, and we all agree that future investigations will continue to build the evidence needed to inform this discrepancy. We designed our paradigm partly with theirs in mind, but we did not replicate their findings. We have so far refrained from framing our data with a focus on the lack of reproducibility of their data, as we favour to offer our data in a neutral context. Our priority (and we assume theirs) is to publish sound data on all sides, for the field (and us) to continue building evidence. It is also essential that the complexity of this divergence be recognised. To begin to address the differences in functional microcircuitry in early vision will require a range of paradigms, animal models, human experiments, and cognitive tasks.

That said, there are a number of points that we can make specifically in relation to our findings and those of Kok et al.

- (1) Analysis differences: We continue to emphasise the methodological differences in both defining the layers and in analysing the functional signals. Our focus is on

differences in information, while their conclusions rest upon univariate responses. We have shown previously (Muckli et al., 2015) that the profile of BOLD amplitudes in cortical depths is not predictive of decoding patterns in the same depth, and we have replicated this using a GRASE sequence, which has higher spatial specificity compared to GE-EPI. Kok et al did not use decoding analyses, so it remains unknown what they would find. These are well documented differences from experts in laminar imaging (e.g. Yang et al., 2021 from the Peter Bandettini group), speaking to several points relevant to this discussion, including that the “complexity of the laminar-specific activity in superficial and deep layers requires further investigation”. We cannot draw too strong conclusions without additional experiments or analyses. However our univariate analysis failed to replicate their findings.

Yang, J., Huber, L., Yu, Y., & Bandettini, P. A. (2021). Linking cortical circuit models to human cognition with laminar fMRI. *Neuroscience & Biobehavioral Reviews*, 128, 467-478.

- (2) Conceptual differences: The authors suggested at the time (2016) that their findings align with predictive coding theory, in as much as “*prediction units being predominantly present in the deep layers and prediction error units in the middle and superficial layers [26]. In the context of the current study, the illusory figure can be seen as a perceptual hypothesis (prediction) and would thus be expected to be encoded by prediction units in the deep layers of V1*”. In collaboration with a group in Amsterdam, we have preliminary evidence in mice that neurons in L2/3 respond with a negative prediction error, in other words when seeing partially occluded images (the same images as in our human study) there is a surprising absence of expected information that neurons in upper layers respond to. This is perceptually similar to our amodal condition where neurons anticipate a contour that is absent. Based on the mice data (which is also consistent with our previous layer fMRI findings (Muckli et al., 2015)), one might hypothesise indeed that illusory contours would also be in superficial layers. The claim in Kok et al., is also well motivated but we stress that there are other interpretations that are emerging. Our broad approach is to test these intuitions in different paradigms and with animal models.
- (3) Stimulus differences: The structure–function organisation of the cortical neurons activated in each of our paradigms may be different. As pointed out by R1, another possible reason for our disparity is that we have colour information in our stimulus, while Kok et al. uses stimulus orientation. How the difference in stimuli could affect V1 processes differently remains an open question. Using orientation selectivity (presumably) takes advantage of the fact that it “is distributed across the whole orientation column”. However, colour processing, too, seems to involve all layers, with colour blobs being present in upper layers, while feedforward and feedback signals that carry red-green colour information are also processed in mid and deep layers (also see response to Reviewer 1, 2nd minor point; and screenshot below of Fig.2 of Briggs & Usrey 2011). Furthermore, electrophysiology research in mice and primates looking at edge perception in figure-ground stimuli also found selective activation of layer 2 & 3, in line with our results. We have added this point of discussion to the manuscript.

We have edited the manuscript as follows:

“Although this superficial layer effect for illusory perception contradicts earlier laminar fMRI findings⁷, it aligns well with results on edge perception between figure and ground from electrophysiology research in mice²⁰ and primates²¹. It should be noted that we use colour stimuli, whereas the contradicting earlier laminar fMRI findings used orientation stimuli. How and whether this may have influenced the results differently remains an open question. Neurons responding to orientation may be distributed throughout the column. However, colour processing, too, appears to involve many layers, with colour blobs being present in upper layers, while feedforward and feedback signals that carry red-green colour information are also processed in mid and deep layers (Briggs & Usrey, 2011).”

Figure 2. Anatomy of feedforward and feedback connections between the LGN and visual cortex (V1)
 A, Nissl-stained section of V1 from the macaque monkey. Corticogeniculate neurones are located exclusively in layer 6. (w.m., white matter.) B, organization of connections between the LGN and V1. The magnocellular layers of the LGN (1 and 2) are shown in grey, the parvocellular layers (3, 4, 5 and 6) are shown in green and red, the koniocellular layers are located below each of the magnocellular and parvocellular layers and are shown in blue. Magnocellular LGN axons (M) terminate in layers 4C α and lower layer 6, parvocellular LGN axons (P) terminate in layers 4C β and upper layer 6, koniocellular LGN axons (K) terminate in layer 4A, the cytochrome oxidase rich blobs and layer 1. The intrinsic connections in V1 maintain the magno- and parvocellular divisions of layers 4C and 6. Neurons in layer 6 of cortex provide feedback to the LGN. Neurons in the upper third of layer 6 project exclusively to the parvocellular LGN layers. Neurons in the lower third of layer 6 project primarily to the magnocellular layers and perhaps the koniocellular layers.

(screenshot from Briggs & Usrey, 2011)

2. The study design is mainly data driven, and the fact that representations can be decoded using data driven approaches leaves open the possibility that encoding may be performed upstream, as actually also pointed out by the authors. Would it be possible for the authors to show decoding from such upstream regions? The authors should address the issue whether these regions can be identified using their data.

We agree that decoding in upstream areas is an exciting avenue to be investigated further. We used a reduced field of view to gain higher spatial resolution, so we are limited to the region from which we acquired fMRI data, only covering the early visual cortex. It is also essential to note that our design is focussed on retinotopic regions that are not stimulated by thalamocortical input. The retinotopic separation of bottom-up (thalamocortical) stimulated regions and those that are not is increasingly difficult in higher visual areas, where population receptive fields are bigger and more likely to overlap.

3. The sample size is not very large and effect sizes are moderate. The authors should provide a justification for sample size and an explicit discussion of power and effect sizes.

We have now performed a new analysis where we find additional statistical support for our claims (see R1 point 1). As outlined above, we have now performed a new linear mixed model analysis to inform on the relative levels of evidence for our effects, that provided further support for our initial analysis, and which we have now added to the manuscript. We think this is the optimal strategy to capture information about the robustness of our effects, and that this probably best aligns also with the reviewer's perspective, and of course, this reviewer didn't have access to the model analysis above prior to making this point. (However, for comparison, we nevertheless provide a further population-level inference analysis here, by computing the population prevalence, a measure of the existence of an effect at the population level. We report here a population prevalence for imagery in the deepest layer (0.9) as 0.27, CI = [0.11 0.47], and for illusion in the superficial layer (0.26) as 0.15, CI = [0.03 0.33]. It is important to also keep in mind that we used selective group of subjects, screened for high imagery abilities – this is a technique that has been proposed to enhance sensitivity when working with relatively small sample sizes (see de Haas, B, 2018. *How to Enhance the Power to Detect Brain–Behavior Correlations With Limited Resources*. *Frontiers in Human Neuroscience* 12, 1–9).

We also highlight that our analysis pipeline requires bespoke meticulous segmentations of grey/white matter in BOLD data. Our Muckli et al., (Current Biology, 2015) data were recorded with 8 subjects and are now replicated in monkeys (Papale et al., Current Biology, 2023) and mice (in prep), all of which give further credit to the reliability of our results and sample size.

For population prevalence see: Ince et al., 2021. *Elife*, <https://elifesciences.org/articles/62461>

For monkey results: Papale et al., 2023. The representation of occluded image regions in area V1 of monkeys and humans. *Current Biology*, <https://doi.org/10.1016/j.cub.2023.08.010>

4. In the 'amodal' version of the stimulus the white contour in the area between the pacman-like rings does not fully break the illusory experience but instead generates the perception of occlusion. This is partly acknowledged but should be further discussed also in terms of the observed experimental results.

The reviewer points to other findings in our lab (Muckli et al, 2015), where we found that occluded natural scene images lead to contextual feedback in the superficial layers of V1. Here we find that the amodal version of the stimulus (in which the coloured squares also appear occluded) is decodable also in the superficial layers. We have now added a sentence

“(C), the ‘amodal’, i.e. occluded version of the illusory stimulus showed significant decoding in the most superficial layer ($p_{adj} = .03$). In separate experiments, we have also observed that occluded natural scene information is found in superficial layers¹⁹.”

5. Regarding the 3 main conditions (imagery, perception, and illusory perception) what would the authors expect if participants would imagine an illusory contour? Although this is not a required experiment they should discuss this in the context of future work.

We hypothesise that this instruction would lead to preferential activation of deep layers, consistent with the profile of mental imagery content that we observe. This is an interesting question and in follow-up experiments, we are currently investigating the V1 profile of imagery for natural scene information, in participants with strong imagery abilities and also in aphantasia. This current study lays the groundwork for these follow up investigations. The preliminary results with natural scenes seem to indicate that amodal completion and modal completion in superficial layers are independent of imagery of such completions.

6. The “imagery” profile in plot in Figure 2C seems relatively flat. What could be the reason for that?

Whilst the imagery profile in Figure 2C appears slightly less steep than in experiment 1, our statistical analyses suggest that this is not meaningful. We have now performed a model comparison and find that our findings do not differ between experiments. In our new linear mixed model analysis mentioned above (R1, point 2), we find no significant main effect of experiment ($t(283) = -1.35, p = 0.178$), no significant interaction between experiment and depth ($t(283) = 1.47, p = 0.143$), or between experiment, condition and depth ($t(283) = -1.43, p = 0.155$), (see figure above Model predictions for individual experiments, now also added to the manuscript as **Supplementary Fig. 13**).

Reviewer #3 (Remarks to the Author):

Bergmann et al. performed sub-mm resolution fMRI in humans during mental imagery, when presented with visual illusions, or when real stimuli were projected on the retina. In striate cortex, they observed differences in activation patterns as a function of cortical depth between these three conditions. Mental imagery can be decoded from fMRI activity in superficial V1 layers, while activity in deeper layers can be used to decode visual illusions. The authors suggest that different microcircuits are recruited during subjective experiences evoked by imagery and stimuli inducing illusions. While the results obtained with the illusory objects confirmed previous research from the same group, it contradicted the results from the group of Floris de Lange.

The reviewer notes that we find different experiential forms of internal experiences localised to different cortical layers in V1, however in the opposite pattern the reviewer describes (a simple oversight we assume - our finding is that imagery is more prominent in deep layers and modal and amodal completion in superficial layers). Throughout this rebuttal we address the question of how our data relates to those of Kok and De Lange (e.g. see more below and R2, point 1).

In general, the study has been executed with great care. The behavioral assessment for the strength of imagination of individual subjects is highly commendable. Unfortunately, this information is not used to explain the inter-individual variability of the fMRI results. The analyses are state of the art. Moreover, it is also commendable that most data of individual subjects are shown (but see some suggestions below).

We thank the reviewer for this positive feedback. The reviewer suggests a very interesting idea to associate behavioural imagery abilities with decoding performance. However, we used a highly specific sample of subjects, screened to confirm they performed over a threshold, saturating any range of variability in behavioural performance (as expected, the results are hence non-significant, see further below). It would indeed be very interesting to examine our findings in aphantasia and hyperphantasia, whereby imagery ability (or lack thereof) might co-vary with brain data.

Although the study is nicely and carefully executed and although I tend to believe the results, I suspect that the ‘real’ functional effects are exceedingly subtle and very hard to capture with fMRI, even using high field scanners. My main reason for my reduced enthusiasm is the high inter-subject variability as shown in the supplementary information. The illusory and mental imagery data are quite variable across subjects, sometimes even showing opposite results or trends. One has to consider that the ‘average’ group result, on which the main conclusions are based, are simply a reflection of the relatively small group of subjects being tested. This may also explain the differences between the present study and that of de Lange’s group. In other words, one should consider the possibility that really large numbers of subjects need to be tested before conclusive statements can be made. Hence the study may be under-powered. A stringent power analysis is warranted in this regard. Also, a comparison of the results of both hemispheres separately for each

subject might provide insightful information. There may be considerable inter-subject variability, certainly concerning the strategies to perform a mental imagery task and the relevant circuitry being employed. However, one does not expect major differences across hemispheres of the same subject. Inter-hemispheric correlations should be higher than inter-subject correlations in decoding strength as a function of laminar depth. The major question is whether 16 subjects is enough to reveal conclusive results.

We absolutely agree that such laminar-specific top-effects are subtle, especially in comparison to sensory-driven, area-wise signals. This is, in part, what makes our data so compelling and important for the field. Taken together, our meticulous analyses, replication across the 2 experiments and ROIs, confirmation with different paradigms in our lab and our consistency with animal data in our broader research agenda are all providing a systematic accumulation of evidence towards multilevel spatiotemporal characteristics underlying processing in early visual microcircuits, under different visual functions. Further, our new analyses outlined above (R1, point 1) provides an additional level of statistical support for our main conclusions that illusory content is significantly decoded only in the superficial cortex while imagery content is significantly decoded only in deep layers of cortex. Please also see R2, point 3 for an estimate of power using population prevalence; we would further like to point out that we pre-selected our participants based on their imagery strength – this is a method that has been proposed to enhance sensitivity when working with small sample sizes (de Haas, 2018). We refrain from testing each hemisphere separately, as this reduces the number of voxels for the classifier, which is expected to reduce the performance of the classifier. However, the relationship between number of voxel and classifier performance is well documented, for example in our previous work: Morgan et al. (2019). Furthermore, please note that the visual field from which the ROI is defined in Experiment 2 is not symmetrical and ‘evenly distributed’ across the two hemifields (it is a hemicircle along one of the illusory boundaries), which could additionally cause variation.

We appreciate the idea of a framework testing individual variability, for example in imagery abilities, and how this relates to microcircuit recruitment. As mentioned above however, we do not have sufficient variation in imagery performance in this dataset to draw such conclusions, as we selected subjects based on them reaching an upper bound in imagery abilities. Nonetheless, we have added an analysis of this further below.

Another major issue I have is that, by definition, the displays during the different conditions was quite different. How can one exclude the possibility that results are explained by these different stimulation conditions. The subjects make eye-movements - which were difficult to measure and not considered. Even small differences in retinal stimulation (in the fovea) will evoke different activity levels independent of the illusory or imagery conditions. Moreover, influences from outside the classical receptive field of the foveal V1 neurons, induced by the differences in the display, might modulate activity independent of the subjective experiences during the different conditions. To really exclude the influence of saccades and extra-classical receptive field phenomena, the stimuli should be much larger. The question is whether this is feasible.

The displays are different but we normalise for these differences with our fixation points, and we now provide eye movement data to respond to the point of how we exclude the influence of saccades. Eye movements would not induce the stereotypical patterns of results across subjects that we observe. In general eye movements make a consistent classification more difficult. Importantly, we would like to point out that we decode green vs. red colour within each condition (imagery, illusory etc). In other words, our main analysis is not decoding the different conditions against each other (i.e. imagery vs. illusory perception etc.). If eye movements explained the results, there

would need to be typical patterns for green illusory perception vs. red illusory perception, red imagery vs green imagery etc. Further, if eye-movements were different for red vs. green, it would not explain the *layer-wise* patterns of results. In the subjects in which we were able to acquire eye tracking data across all runs we can confirm that those subjects were fixating well. The figure below shows high correlations between eye positions for red and green conditions during both the illusory and imagery states. We have added this as a new supplementary figure.

Along the same lines, why is a faint grey contour shown during the mental imagery condition? A 3.5 deg radius circle may be large enough to evoke activity in foveal representations, certainly when subjects make eye movements.

Is an ISI of 7-8 seconds enough to clear after-images following the 'real' perceptual and the illusory contour conditions?

The faint grey circle helped subjects to localise the imagery content in visual space systematically across trials. The decoding is made between the imagined colours green v red. The faint line is grey so even if it would induce residual activation (which our retinotopic mapping makes sure it doesn't) it would be independent of the imagined colour.

Yes, we designed the paradigm considering that a minimum of 6s would be sufficient. The sequence is randomised so even if perceptual activity would linger in the activation patterns it would be orthogonal to the classification and if anything increase noise level for the classifier.

The behavioral pre-test for mental imagery is commendable. However, is there a correlation between the score and the fMRI data? 32 hemispheres should be sufficient to get a reliable estimate.

This is a very clever idea and would be interesting to examine in a subject cohort with sufficient variability in behavioural performance. However, we used the behavioural pre-test to specifically select individuals with high mental imagery ability, i.e. our cohort were at ceiling performance levels. Due to this selection, a correlation between the score and the fMRI data is less likely compared to if we had used individuals who cover the whole range of imagery abilities. As expected, when pooled across the two experiments, the Spearman rank correlation between the behavioural imagery strength (%primed) and decoding accuracy were non-significant and near

zero - in the deepest layer (0.9), it was $r_s = -.05, p = .84$. At the second deepest layer, the Spearman rank correlation was $r_s = -.13, p = .56$.

Please also note that we refrain from using the hemispheres separately in this analysis, as we only have one behavioural imagery strength value (%primed) per subject. An analysis using all hemispheres separately would only make sense if we also had a separate behavioural imagery strength value per hemisphere (e.g. imagery tested separately in the left and right visual field).

Behavioural imagery strength (pre-test) and fMRI decoding performance. Individual imagery strength and fMRI decoding accuracy at the two deepest cortical depth layers (0.9 and 0.74). We selected our fMRI participants based on their imagery strength measured in a behavioural pre-test. As a consequence, the range of imagery abilities in our cohort was strongly limited, reducing the likelihood that a relationship between decoding performance and behavioural imagery strength would be present. As expected, we only found non-significant Spearman rank correlations. We pooled the subjects from both experiments here. The blue circles represent individual subject data. For the four participants which had taken part in both fMRI experiments, decoding performance was averaged across the two sessions before correlating it with behaviour.

Did the authors consider a decoding analysis as a function of eccentricity & laminar depth?

Experiment 1 was decoding in more peripheral regions than Experiment 2. Separating eccentric subsections in the ROIs is an interesting suggestion but is reaching limitations in power as rightly pointed out in reviewers comments above.

However, one analysis that we conducted which may be of interest to the reviewer in this regard is the one using visual projections, see Suppl. Figure 4C. Here, we looked at the relationship between proximity to the inducer of the visual field portion that each voxel of the ROI represents, and that visual field portion's influence on the SVM classification analysis (separately for each depth layer). The results do not suggest that visual field locations closer to the inducers are more influential in the classification analysis.

Furthermore, we have also tested the influence of eccentricity in decoding amodal completion of natural scenes in a previous publication (Morgan 2019). There we systematically reduced the size of the occluded ROI and repeated classification analyses for individual scenes by shifting the border of the ROI by 0.25° visual angle (up to 2.75°) to further points of eccentricity. Importantly, while classification performance decreased as we moved away from the image border, this decrease was not statistically different from a general decrease in ROI size. This result shows that the observed decrease in performance is related to the number of voxels included, not to their proximity to (in that case) the occluder border.

Line 385: Fig 2A \diamond 1A

We have amended this typo on line 385.

Line 577: It is unclear why the analyses jump from 3D to 2D to 3D. Staying within 3D should reduce errors, no?

Our analysis remains in 3D. 3D-fMRI voxels are assigned to depth layers. The depth layers are identified by reconstructing the cortical surface and defining the outer pial surface and the white-matter grey-matter boundary. This is the only way with which a 3D-voxel can be assigned to its cortical depth origin.

I'm somewhat puzzled by the segregation between some 'parts' of the same ROI (e.g. S11 in Fig 3). One would expect a continuous uninterrupted ROI -unless it is an illusion induced by the 2D section. It would be highly informative if the ROIs for all subjects are shown in this format and also in flattened 2D format.

The nature of the folding of the visual cortex leads to the segmentation to different sections of a ROI with a 2D plane. V1 is one of the most complex folded regions in human cortex. In the inflated cortical reconstruction it is a continuous well formed structure. In the original folded brain, sections of one ROI can appear disjunct.

Line 650: A potential explanation for the differences between the current study and Kok et al., is that the latter looked at univariate activity differences evoked by an illusory contour versus nothing (while in the present study SVMs were used to decode two illusory contours from each other). However, to resolve the issue, the authors could also perform the same univariate analyses as Kok et al.

Yes this is a very important point. We have addressed this point further above in rebuttal R1 and R2 and have added additional analyses to directly compare the univariate results in the conditions in question to the manuscript (see caption in Supplementary Fig. 6; illusory against mock, illusory against amodal; separately for each colour).

In sum, we thank all reviewers again for their attention and outstanding level of care and focus.

Reviewer #1 (Remarks to the Author):

The authors have fully addressed most of my concerns, but a few were addressed only partially/tangentially.

To address major points 1 and 2 the authors now provide the results of the linear mixed model analysis, combining the data of both experiments. While using more data is a sensible approach, as I pointed out in major point 2, experiment 2 where the same voxels are exposed to different experimental conditions is the most informative in my opinion. When this experiment is considered separately, there is no evidence for a double-dissociation between layers and processes (similar above-chance decoding in the superficial layers is observed for both, imagery and illusory perception). The mixed model analysis the authors provide does not show a significant effect of experiment and no interaction between depth and experiment, but based on classical inferential statistics we cannot claim that these effects are absent. I would insist that the authors at least acknowledge that the results in the superficial layers are less conclusive and need further investigation in the *main* manuscript.

In addressing my minor point 1, the authors added a mention of the study by Iamshchinina et al. and its main finding, but failed to provide an explanation why that study could find above-chance decoding of imagery when trained on perception, but the current study could not. A brief statement on the reasons for divergence between the current and Iamshchinina et al. findings with regard to cross-decoding (perception -> imagery) in the manuscript is necessary in my opinion.

Supplementary materials Line 328: There is a missing horizontal arrow in the supplementary Figure 13 below the x-axes of the left plot.

Reviewer #3 (Remarks to the Author):

The authors did an excellent job addressing the comments of all reviewers. As other reviewers and I stated in the first reviewing round, the effect size is very subtle -which is exactly what one would have predicted.

I don't have any further questions and commend the authors for this very nice piece of work

REVIEWERS' COMMENTS

Reviewer #1 (Remarks to the Author):

The authors have fully addressed most of my concerns, but a few were addressed only partially/tangentially.

We wish to thank R1 again for exceptional attention in providing feedback, and we're happy to address these last two important points.

To address major points 1 and 2 the authors now provide the results of the linear mixed model analysis, combining the data of both experiments. While using more data is a sensible approach, as I pointed out in major point 2, experiment 2 where the same voxels are exposed to different experimental conditions is the most informative in my opinion. When this experiment is considered separately, there is no evidence for a double-dissociation between layers and processes (similar above-chance decoding in the superficial layers is observed for both, imagery and illusory perception). The mixed model analysis the authors provide does not show a significant effect of experiment and no interaction between depth and experiment, but based on classical inferential statistics we cannot claim that these effects are absent. I would insist that the authors at least acknowledge that the results in the superficial layers are less conclusive and need further investigation in the *main* manuscript.

We have added this point for clarification. We have now edited the main manuscript as follows:

“There was no significant main effect of experiment ($t(283) = -1.35$, $p = 0.178$, $\beta = -0.07$, 95% CI [-0.17, 0.03]) and no significant interactions between experiment and depth ($t(283) = 1.47$, $p = 0.143$, $\beta = 0.14$, 95% CI [-0.05, 0.33]), or between experiment, condition and depth ($t(283) = -1.43$, $p = 0.155$, $\beta = -0.17$, 95% CI [-0.40, 0.06]), together indicating that the experiment did not significantly affect the slope. In conclusion, the interaction between condition and cortical depth indicates that decoding significantly differed between the critical experimental conditions (mental imagery and illusory perception) at the different cortical depths. This analysis complements the first-level bootstrapping analyses showing that illusory content is significantly decoded only in the superficial cortical depths while imagery content is significantly decoded only at deeper cortical depths. However, the absence of significant decoding of imagery in the superficial layers is not evidence for no decoding in superficial layers. Note that we did see individual participants with decoding of imagery content in the superficial layers, (especially in experiment 2, but also in some participants in experiment 1, **Supplementary Fig. 10 and 12**). Further investigations might be able to link decoding fluctuations in superficial layers to the subjective strength of imagery experienced (see discussion)”.

(For reference, this is what we already say in the discussion:

*Interestingly, although overall decoding of mental imagery remained non-significant in superficial layers, there were some individuals in our experiments who showed above-chance decoding here, too, whereas others did not (**Supplementary Fig. 10 and 12**). It is possible that inter- (and intra-) individual differences in the laminar information profile of mental imagery might account for how strong or ‘real’ it might be experienced: that is, a mental imagery experience might appear stronger, more*

precise or more 'real' when a larger portion of imagery-related signals reach not only deep, but also superficial layers via the longer/slower path through the visual cortex hierarchy, or through bifurcations from the deep layers).

In addressing my minor point 1, the authors added a mention of the study by lamshchinina et al. and its main finding, but failed to provide an explanation why that study could find above-chance decoding of imagery when trained on perception, but the current study could not. A brief statement on the reasons for divergence between the current and lamshchinina et al. findings with regard to cross-decoding (perception -> imagery) in the manuscript is necessary in my opinion.

Thanks for pointing this out. We had addressed this point in the rebuttal, and we agree that it is important to make this explicit in the manuscript, too. We made an amendment to now more thoroughly address this point.

Main manuscript:

“Interestingly, when training the classifier on illusory colour and testing it on perceptual colour, we found that significant above-chance decoding was again only present in superficial layers, suggesting that information between illusory and actual perception is shared at these depths (**Supplementary Fig. 5**). In contrast, we could neither cross-classify between mental imagery and illusory perception, nor between mental imagery and perception, irrespective of whether we trained on mental imagery and tested on (illusory) perception or vice versa. This suggests that V1 feedback information of mental imagery may be more distinct. However, it has been shown previously that a classifier trained on response patterns when participants saw oriented gratings could be successfully tested on response patterns when participants were mentally rotating the gratings²⁴. This discrepancy requires follow up investigations but could plausibly relate to the fact that the processing of static imagery, as in our experiment, might be different to the mental rotation task used previously. Evidence in support of this hypothesis comes from congenital aphantasia where participants who are unable to form visual mental images perform with similar accuracy as controls in mental rotation tasks²⁵.”

Supplementary Figure 5:

“When using smaller voxels and decoding for each layer separately, voxels are less prone to pooling signals over a larger space and from a larger range of cortical depths. In this case, it may be that the (layer-wise) divergences between these two experiences become more apparent. However, with layer resolution data, lamshchinina et al.²⁴ found cross-classification effects in a mental rotation task (see main manuscript for discussion).

Supplementary materials Line 328: There is a missing horizontal arrow in the supplementary Figure 13 below the x-axes of the left plot.

Thanks for making us aware of this - we have added this arrow.

Reviewer #3 (Remarks to the Author):

The authors did an excellent job addressing the comments of all reviewers. As other reviewers and I stated in the first reviewing round, the effect size is very subtle - which is exactly what one would have predicted. I don't have any further questions and commend the authors for this very nice piece of work

We thank R3 once again for their excellent input that helped us greatly.